# D2 Actor Critic:
# Diffusion Actor Meets Distributional Critic

**Lunjun Zhang** *                                              *lunjun@cs.toronto.edu*
*Department of Computer Science, University of Toronto*

**Shuo Han** *                                                 *TansioHan@u.northwestern.edu*
*Department of Statistics, Northwestern University*

**Hanrui Lyu**                                                 *HanruiLyu2029@u.northwestern.edu*
*Department of Statistics, Northwestern University*

**Bradly C Stadie**                                           *bradly.stadie@northwestern.edu*
*Department of Statistics, Northwestern University*

**Reviewed on OpenReview:** *https://openreview.net/forum?id=8KbstCUXhH*

## Abstract

We introduce **D2AC**, a new model-free reinforcement learning (RL) algorithm designed to train expressive diffusion policies online effectively. At its core is a policy improvement objective that avoids the high variance of typical policy gradients and the complexity of backpropagation through time. This stable learning process is critically enabled by our second contribution: a robust distributional critic, which we design through a fusion of distributional RL and clipped double Q-learning. The resulting algorithm is highly effective, achieving state-of-the-art performance on a benchmark of eighteen hard RL tasks, including Humanoid, Dog, and Shadow Hand domains, spanning both dense-reward and goal-conditioned RL scenarios. Beyond standard benchmarks, we also evaluate a biologically motivated predator-prey task to examine the behavioral robustness and generalization capacity of our approach.

## 1 Introduction

Actor-Critic methods have been at the center of many recent advances in Reinforcement Learning. In continuous control and robotics, Soft Actor-Critic (Haarnoja et al., 2018) outperformed strong prior model-based methods (Wang et al., 2019). On the ALE environment, Atari agents such as Rainbow (Hessel et al., 2018), BBF (Schwarzer et al., 2023), and ULTHO (Yuan et al., 2025) have continued to push forward state-of-the-art reinforcement learning performance.

Each instantiation of actor-critic methods crucially relies on two components: an actor and a critic. The critic tells us how valuable it is to be in a given state, and the actor tells us what we should do when we find ourselves in that state. Generally, an actor-critic algorithm will only be as strong as the weakest link; a powerful critic is wasted on an actor that can't optimize for the correct actions, and vice versa. This naturally leads us to ask the question of how we can use modern advancements in machine learning and function estimation to learn a critic and an actor that are a perfect match, equally powerful, and capable of complementing one another.

When estimating the critic, classical RL favored point estimates for the value function, often resulting in unstable estimation. While modeling the full distribution of returns (Bellemare et al., 2017) significantly improves robustness—a principle reinforced by a recent trend advocating for classification-style objectives

---

*Equal contribution.

over standard regression for value learning (Imani et al., 2024; Farebrother et al., 2024)—we observe that this approach is not immune to the overestimation bias inherent in Q-learning. Our first contribution is to create a more stable teacher for the actor: we find that applying the clipped double Q-learning technique from TD3 (Fujimoto et al., 2018) to approximate the return distribution reduces noise and provides a more reliable learning signal for policy improvement.

However, even a stable teacher is wasted on an actor that cannot effectively utilize its guidance. While recent work has explored powerful denoising diffusion models for the policy (Yang et al., 2023), a core challenge remains: how to effectively bridge the critic's guidance with the actor's learning process? Policy gradient estimation for diffusion policies often suffers from high variance when applied to diffusion models (Jeha et al., 2024), while other methods can be complex and unstable (Ding et al., 2024; Ma et al., 2025).

This brings us to the core technical contribution of our work. We derive a new policy improvement objective for diffusion actors that is both theoretically grounded and practically efficient. Taking inspiration from monotonic policy improvement theory (Schulman et al., 2015), our derivation simplifies the complex, multi-step policy optimization into a stable, single-step supervised objective guided directly by the critic's value gradients. This simplified objective provides the crucial link, allowing our expressive diffusion actor to effectively learn from our stable distributional critic (i.e., a critic that models the full distribution of returns), forming a more synergistic actor-critic system.

In this paper, we show how to marry a distributional critic to a diffusion actor, leading to the development of D2AC, a new actor-critic algorithm. D2AC achieves excellent performance on a variety of hard robotics environments, including locomotion tasks such as Humanoid and Dog domains in DeepMind Control Suite (Tassa et al., 2018), and manipulation tasks such as Pick-and-Place and Shadow-Hand Manipulate in multi-goal RL environments with sparse rewards (Plappert et al., 2018). In addition, we evaluate D2AC on a biology-inspired predator–prey benchmark (Lai et al., 2024), where it demonstrates higher exploration coverage, richer path diversity, and superior zero-shot transfer compared to SAC and TD-MPC2. Importantly, D2AC consistently performs well across difficult domains where prior model-free RL methods struggle or completely fail. Moreover, it significantly closes the performance gap between model-free RL and SOTA model-based methods such as TD-MPC2 (Hansen et al., 2023), despite using an order-of-magnitude less compute compared to TD-MPC2 and being considerably faster to run. Our results show that D2AC can serve as a strong base for policy optimization in continuous control. Videos of our policy in action can be found here [1].

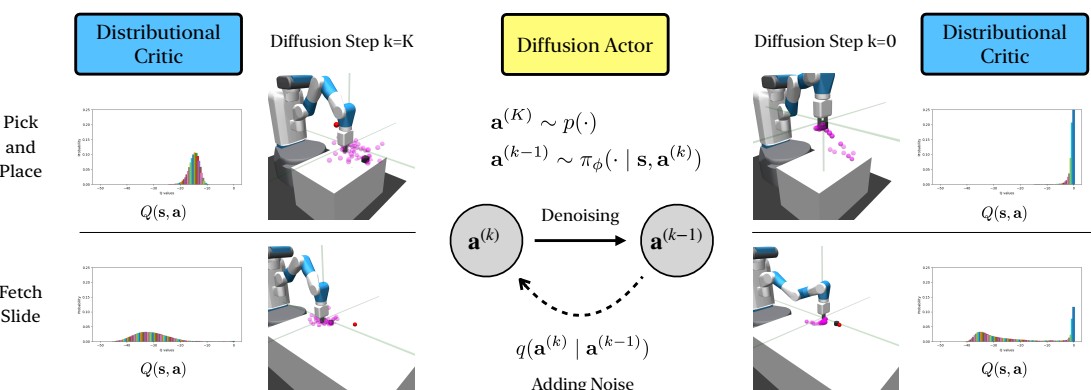

Figure 1: **D2 Actor Critic** uses a critic that models a distribution over possible returns. A diffusion actor uses the expected value of this distribution (the Q-function) to help align the denoising process with policy improvement. Above are visualizations of the Pick-and-Place and Fetch Slide environments.

---

[1] https://d2ac-actor-critic.github.io/

## 2 Related Works

**Model-free RL** has two families of approaches: policy gradient methods and value-based methods. Policy gradient methods typically require on-policy data and trust-region regularization (Schulman et al., 2015; 2017b). Value-based methods, on the other hand, enable off-policy training, which can greatly improve sample efficiency. Since DQN (Mnih et al., 2015), actor-critic extensions like DDPG (Lillicrap et al., 2015) brought these methods to continuous control, and subsequent improvements include tackling Q-value overestimation in TD3 (Fujimoto et al., 2018) and the introduction of soft Q-learning (Haarnoja et al., 2018). A particularly relevant line of work is distributional RL (Bellemare et al., 2017), which moves beyond point-estimates to model the entire distribution of returns. This concept has been successfully integrated into actor-critic frameworks to address value estimation errors or enable risk-sensitive learning, as demonstrated by some concurrent works like DSAC (Ma et al., 2025). D2AC builds directly on this progress, unifying a robust distributional critic with clipped Q-learning and, crucially, introducing a novel actor update mechanism designed to leverage these richer value representations.

**Model-based RL and Planning:** The paradigm of learning a world model and planning with it has achieved significant breakthroughs (Schrittwieser et al., 2020; Hansen et al., 2022). We are particularly inspired by core mechanisms of modern planners like Model Predictive Path Integral Control (MPPI) (Williams et al., 2015), where action proposals are iteratively refined based on insights from a full distribution of simulated future returns. D2AC is an attempt to distill these principles into a cohesive, purely model-free framework, making our approach orthogonal to and potentially a stronger base for future hybrid methods.

**Diffusion Policy for RL** has attracted interest following the success of denoising diffusion models (Sohl-Dickstein et al., 2015; Ho et al., 2020; Song et al., 2020b). The most straightforward application is conditional behavior cloning (BC) (Janner et al., 2022; Chi et al., 2023; Römer et al., 2024). Extending diffusion policies beyond BC, prior works have explored both offline RL (Hansen-Estruch et al., 2023; Gao et al., 2025) and online RL settings (Chen et al., 2023; Ma et al., 2025; Dong et al., 2025). A key challenge, however, is that policy improvement in online RL necessarily induces a non-stationary target distribution, which makes diffusion-based training unstable. Our work takes a different perspective: drawing on monotonic improvement theory from policy optimization literature (Schulman et al., 2015), we derive a new method for training diffusion policies in the online setting. This is different from some concurrent approaches, such as Liu et al. (2025), which also combine a distributional critic with a diffusion policy but rely on policy gradient updates. Our core technical contribution thus offers a distinct and efficient alternative for actor training in this area. We note that Liu et al. (2025) was developed concurrently and view our work as complementary in exploring different optimization perspectives.

## 3 Background

### 3.1 Reinforcement Learning

A Markov Decision Process (MDP) is defined by the state space $\mathcal{S}$, action space $\mathcal{A}$, the initial state distribution $\rho_0(\mathbf{s})$, the transition probability $p(\mathbf{s}' \mid \mathbf{s}, \mathbf{a})$, and the reward function $r(\mathbf{s}, \mathbf{a})$.

The policy $\pi(\mathbf{a} \mid \mathbf{s})$ generates a trajectory $\tau$ with horizon $T$ within the environment:

$$\tau = \{\mathbf{s}_0, \mathbf{a}_0, \cdots, \mathbf{s}_T\} \tag{1}$$

where $\mathbf{s}_0 \sim \rho_0(\cdot)$, $\mathbf{a}_t \sim \pi(\cdot \mid \mathbf{s}_t)$, and $\mathbf{s}_{t+1} \sim p(\cdot \mid \mathbf{s}_t, \mathbf{a}_t)$ for $t \in \{0, \cdots, T-1\}$.

We denote the trajectory distribution starting from state $s$ under policy $\pi$ as $p_\pi(\tau \mid \mathbf{s})$. With a discount factor $\gamma \in (0, 1)$, the goal is for the policy $\pi$ to maximize the discounted cumulative rewards:

$$\mathcal{J}(\pi) = \mathbb{E}_{\rho_0(\mathbf{s}_0) p_\pi(\tau \mid \mathbf{s}_0)} \left[ \sum_{t=0}^{T-1} \gamma^t r(\mathbf{s}_t, \mathbf{a}_t) \right] \tag{2}$$

Value-based RL methods optimize this objective by estimating a Q function for the current policy $\pi$:

$$Q(\mathbf{s}_t, \mathbf{a}_t) = \mathbb{E}_{p(\mathbf{s}_{t+1}|\mathbf{s}_t, \mathbf{a}_t)p_\pi(\tau|\mathbf{s}_{t+1})} \left[ \sum_{\Delta=0}^{T-t} \gamma^\Delta r(\mathbf{s}_{t+\Delta}, \mathbf{a}_{t+\Delta}) \right] \tag{3}$$

and then training the policy $\pi$ to maximize Q.

### 3.2 Denoising Diffusion Models

Denoising Diffusion Models are generative models that estimate the gradients of data distribution (Vincent, 2011; Song & Ermon, 2019; Song et al., 2020a). Diffusion models train a denoiser $D_\phi$ to reconstruct data $\mathbf{x} \sim p_{\text{data}}$ from noised data $\mathbf{x}^{(k)} \sim \mathcal{N}(\mathbf{x}, \sigma_k^2 \mathbf{I})$ under various noise levels $\sigma_{\max} = \sigma_K > \cdots > \sigma_0 = \sigma_{\min}$.

Let the distribution of noised data be defined as:

$$p(\mathbf{x}^{(k)}) = \int p_{\text{data}}(\mathbf{x}) \mathcal{N}(\mathbf{x}^{(k)}|\mathbf{x}, \sigma_k^2 \mathbf{I}) d\mathbf{x} \tag{4}$$

The min $\sigma_{\min}$ and max $\sigma_{\max}$ of noise levels are selected such that $p(\mathbf{x}^{(0)}) \approx p_{\text{data}}(\mathbf{x})$ and $p(\mathbf{x}^{(K)}) \approx \mathcal{N}(\mathbf{0}, \sigma_{\max}^2 \mathbf{I})$.

The diffusion loss is typically:

$$\mathbb{E}_{\boldsymbol{\epsilon} \sim \mathcal{N}(\mathbf{0}, \mathbf{I}), \, k \sim \text{Unif}(1 \cdots K)} [\lambda(k) \| D_\phi(\mathbf{x} + \boldsymbol{\epsilon}\sigma_k; \sigma_k) - \mathbf{x} \|^2] \tag{5}$$

where $\lambda(k)$ is the loss weighting based on noise level.

Then the score for arbitrary input $\mathbf{x}$ is:

$$\nabla_{\mathbf{x}} \log p(\mathbf{x}; \sigma) = \frac{D_\phi(\mathbf{x}; \sigma) - \mathbf{x}}{\sigma^2} \tag{6}$$

which can be used to sample data by starting from noise and solving either an SDE or ODE (Song et al., 2020b).

We follow EDM (Karras et al., 2022) to define the denoiser as:

$$D_\phi(\mathbf{x}, \sigma) = c_{\text{skip}}(\sigma)\mathbf{x} + c_{\text{out}}(\sigma)F_\phi(c_{\text{in}}(\sigma)\mathbf{x}, c_{\text{noise}}(\sigma)) \tag{7}$$

where $F_\phi$ is a neural network, and $c_{\text{skip}}(\sigma), c_{\text{out}}(\sigma), c_{\text{in}}(\sigma), c_{\text{noise}}(\sigma)$ are $\sigma$-dependent scaling coefficients defined in (Karras et al., 2022). These coefficients control the relative weighting of the input signal, the output of the neural network, and the noise embedding across different noise levels.

### 3.3 Clipped Double Q-learning for Categorical Return Distributions

While distributional reinforcement learning (RL) provides richer return modeling by capturing full return distributions instead of scalar estimates (Bellemare et al., 2017), it inherits a well-known flaw of traditional Q-learning: overestimation bias (Van Hasselt, 2010; Van Hasselt et al., 2016). In distributional RL, this bias manifests not just in the expected value, but across entire regions of the predicted return distribution, leading to pathological optimism.

To address this, we apply a clipped double Q-learning approach (Fujimoto et al., 2018) within the distributional setting. Though this combination is conceptually simple, we demonstrate in Figure 9 that it plays a crucial role in stabilizing training and improving empirical performance.

**Categorical Representation of Return Distributions:**  We represent return distributions using the categorical formulation introduced in Bellemare et al. (2017). Specifically, our distributional model learns to predict a probability distribution, $\mathbf{q}_\theta(\mathbf{s}, \mathbf{a})$, over a fixed, discrete support of possible returns $\mathbf{z}_q =$

$[V_{\min}, \ldots, V_{\max}]$. This distribution is parameterized by a softmax output over $N + 1$ bins, where the probability of each bin corresponds to an atom in the support $\mathbf{z}$. The expected Q-value is computed as:

$$Q_\theta(\mathbf{s}, \mathbf{a}) = \mathbf{q}_\theta(\mathbf{s}, \mathbf{a})^\top \mathbf{z}_q$$

Any continuous value within the support range can be represented via a linear combination of two adjacent bins using a *two-hot* encoding function $h_{\mathbf{z}_q}(\cdot)$.

**Categorical Projection:** To perform bootstrapped learning, we shift the support according to the observed reward and discount factor: $\mathbf{z}_p = r + \gamma \mathbf{z}_q$. The predicted distribution $\mathbf{p} = \mathbf{q}_{\bar{\theta}}(\mathbf{s}', \mathbf{a}')$ under this shifted support must be projected back to the original support $\mathbf{z}_q$ to compute a training target. This is done via:

$$\mathbf{\Phi}_{\text{dist}}(\mathbf{z}_p, \mathbf{p}, \mathbf{z}_q)[k] = \sum_{j=0}^{N} h_{\mathbf{z}_q}(\mathbf{z}_p[j])[k] \cdot \mathbf{p}[j]$$

This yields a valid probability distribution over the original support, which serves as a supervised signal.

**Clipped Double Distributional RL:** To reduce overestimation, we maintain two independent critics, $\mathbf{q}_{\theta_1}$ and $\mathbf{q}_{\theta_2}$, each producing a distribution over returns. For each training step, we compute expected Q-values:

$$Q_{\theta_i}(\mathbf{s}, \mathbf{a}) = \mathbf{q}_{\theta_i}(\mathbf{s}, \mathbf{a})^\top \mathbf{z}_q$$

We then select the entire distribution from the critic with the lower expected value:

$$\mathbf{q}_\theta^{\text{clip}}(\mathbf{s}, \mathbf{a}) = \begin{cases} \mathbf{q}_{\theta_1}(\mathbf{s}, \mathbf{a}) & \text{if } Q_{\theta_1} \leq Q_{\theta_2} \\ \mathbf{q}_{\theta_2}(\mathbf{s}, \mathbf{a}) & \text{otherwise} \end{cases}$$

This clipped distribution is projected and used to train both critics via cross-entropy loss. The full critic loss becomes:

$$L_{\text{critic}}(\theta) = -\boldsymbol{\ell}^\top \left( \log \mathbf{q}_{\theta_1}(\mathbf{s}, \mathbf{a}) + \log \mathbf{q}_{\theta_2}(\mathbf{s}, \mathbf{a}) \right)$$

where $\boldsymbol{\ell} = \mathbf{\Phi}_{\text{dist}}(r + \gamma \mathbf{z}_q, \mathbf{q}_{\bar{\theta}}^{\text{clip}}(\mathbf{s}', \mathbf{a}'), \mathbf{z}_q)$ and $\mathbf{a}' \sim \pi_\phi(\cdot \mid \mathbf{s}')$.

## 4 Policy Improvement for Diffusion Actors

This section introduces the core technical contribution of our work: an efficient policy improvement objective for diffusion actors. To derive this objective, we first reframe the multi-step denoising process as a special type of Markov Decision Process. We then leverage the principles of monotonic policy improvement (Schulman et al., 2015) and, through a key theoretical simplification, arrive at a tractable objective that aligns the single-step denoising function with policy improvement.

When the policy $\pi$ in actor-critic algorithms is a diffusion model (Sohl-Dickstein et al., 2015; Ho et al., 2020), it starts from the noise distribution $\mathcal{N}(0, \sigma_{\max}^2 \mathbf{I})$ and goes through $K$ diffusion steps $\mathbf{a}^{(K)} \to \mathbf{a}^{(K-1)} \cdots \to \mathbf{a}^{(0)}$ using a sequence of noise levels $\sigma_{\max} = \sigma_K > \cdots > \sigma_1 > \sigma_0 = \sigma_{\min}$ to sample an action. If we assume a sampler based on the Euler-Maruyama method (Song et al., 2020b), then one step of the diffusion process $\mathbf{a}^{(k)} \to \mathbf{a}^{(k-1)}$ for policy $\pi_\phi$ can be written as:

$$\pi_\phi(\mathbf{a}^{(k-1)} \mid \mathbf{s}, \mathbf{a}^{(k)}) = \mathcal{N}(\mathbf{a}^{(k-1)} \mid \boldsymbol{\mu}_\phi(\mathbf{a}^{(k)}, k; \mathbf{s}), (\sigma_k^2 - \sigma_{k-1}^2)\mathbf{I})$$

Where $\boldsymbol{\mu}_\phi$ is a learned denoising process that takes actions away from the noise and towards the desired action distribution. It is parameterized as:

$$\boldsymbol{\mu}_\phi(\mathbf{a}^{(k)}, k; \mathbf{s}) = \mathbf{a}^{(k)} + (\sigma_k^2 - \sigma_{k-1}^2)\nabla_\mathbf{a} \log p(\mathbf{a}^{(k)} \mid \sigma_k, \mathbf{s})$$
$$\nabla_\mathbf{a} \log p(\mathbf{a} \mid \sigma, \mathbf{s}) = (D_\phi(\mathbf{a}, \sigma; \mathbf{s}) - \mathbf{a})/\sigma^2$$

where $D_\phi(\mathbf{a}, \sigma; \mathbf{s}) = c_{\text{skip}}(\sigma)\mathbf{a} + c_{\text{out}}(\sigma)F_\phi(\mathbf{s}, c_{\text{in}}(\sigma)\mathbf{a}, c_{\text{noise}}(\sigma))$. The functions $c_{\text{skip}}(\sigma)$, $c_{\text{out}}(\sigma)$, and $c_{\text{noise}}(\sigma)$ are flexible design choices (Karras et al., 2022). Thus, taking an action $\mathbf{a}^{(0)}$ from the denoising process $\mathbf{a}^{(k)} \to \cdots \mathbf{a}^{(0)}$ can be written as an integral over the denoising steps:

$$\pi_\phi(\mathbf{a} \mid \mathbf{s}) = \int p(\mathbf{a}^{(K)}) \prod_{k=1}^{K} \pi_\phi(\mathbf{a}^{(k-1)} \mid \mathbf{s}, \mathbf{a}^{(k)}) \mathrm{d}\mathbf{a}_{1:K} \tag{8}$$

We would like a way to guarantee the diffusion process actually provides us with a better action. Define the optimal policy to be $p^*(\mathbf{a} \mid \mathbf{s}) = (\exp(Q(\mathbf{s}, \mathbf{a})/\alpha))/Z(\mathbf{s})$, where $\alpha$ is the temperature. The goal of entropy-regularized policy improvement is to minimize the following (Schulman et al., 2017a; Haarnoja et al., 2018; Black et al., 2023):

$$\arg\max_\phi - D_{KL}(\pi_\phi(\mathbf{a} \mid \mathbf{s}) \parallel p^*(\mathbf{a} \mid \mathbf{s})) = \mathbb{E}_{\mathbf{a} \sim \pi_\phi(\cdot \mid \mathbf{s})}[Q(\mathbf{s}, \mathbf{a})] + \alpha H(\pi_\phi(\mathbf{a} \mid \mathbf{s})) \tag{9}$$

**Diffusion Policy Gradients:** In the case of diffusion policies, our main challenge with optimizing for policy improvement is that accessing $\mathbf{a} \sim \pi_\phi$ requires $K$ intermediate sampling steps. To optimize a lower bound on the policy gradient objective, the variational lower bound in Kingma et al. (2021) can be applied with a positive transformation of the Q-values (Peters & Schaal, 2007; Abdolmaleki et al., 2018), with $\lambda_{\text{pg}}(k) = (1/\sigma_{k-1}^2 - 1/\sigma_k^2) \cdot (K/2)$:

$$\mathbb{E}_\pi \mathbb{E}_{\boldsymbol{\epsilon} \sim \mathcal{N}(0, \mathbf{I}), k \sim \text{Unif}(1 \cdots K)}[\lambda_{\text{pg}}(k)\|\mathbf{a} - D_\phi(\mathbf{a} + \sigma_k \boldsymbol{\epsilon}, \sigma_k; \mathbf{s})\|^2 \cdot \exp Q(\mathbf{s}, \mathbf{a})] \tag{10}$$

However, policy gradients tend to suffer from high variance (Jeha et al., 2024), while value gradients (Heess et al., 2015) for diffusion require backpropagation through time (BPTT), which is notoriously difficult (Pascanu et al., 2013; Wallace et al., 2023; Black et al., 2023). Next, we will show that by re-examining the diffusion process through the lens of an MDP, we can circumvent these challenges.

**Diffusion MDP:** To ground our approach in established RL theory, we first define a Diffusion MDP as taking $K$ steps of actions from $p(\mathbf{a}^{(K)})$ to $\pi(\mathbf{a}^{(k-1)} \mid \mathbf{s}, \mathbf{a}^{(k)})$, reaching the final state $\mathbf{a}^{(0)}$. After this diffusion process (under discount factor $\gamma$), the policy receives a reward $Q(\mathbf{s}, \mathbf{a}^{(0)})/\gamma^K$ at the final timestep of diffusion (reward is 0 elsewhere). The objective is to maximize the expected terminal reward:

$$\eta(\mathbf{s}, \pi) = \mathbb{E}_{\mathbf{a}^{(K)} \cdots \mathbf{a}^{(0)} \sim \pi}[Q(\mathbf{s}, \mathbf{a}^{(0)})] \tag{11}$$

This objective makes the dependence on the intermediate diffusion steps $\mathbf{a}^{(K)} \cdots \mathbf{a}^{(1)}$ explicit. Correspondingly, in the diffusion MDP, the Q-function, value function, and advantage function are given by:

$$V_\pi^{\text{diffusion}}((\mathbf{s}, \mathbf{a}^{(k)})) \triangleq \mathbb{E}_{\mathbf{a}^{(k-1)} \cdots \mathbf{a}^{(0)} \sim \pi}[(\gamma^{k-1}/\gamma^K)Q(\mathbf{s}, \mathbf{a}^{(0)})]$$
$$Q_\pi^{\text{diffusion}}((\mathbf{s}, \mathbf{a}^{(k+1)}), \mathbf{a}^{(k)}) \triangleq \mathbb{E}_{\mathbf{a}^{(k-1)} \sim \pi}[\gamma V_\pi^{\text{diffusion}}((\mathbf{s}, \mathbf{a}^{(k-1)}))]$$
$$A_\pi^{\text{diffusion}}((\mathbf{s}, \mathbf{a}^{(k+1)}), \mathbf{a}^{(k)}) \triangleq Q_\pi^{\text{diffusion}}((\mathbf{s}, \mathbf{a}^{(k+1)}), \mathbf{a}^{(k)}) - V_\pi^{\text{diffusion}}((\mathbf{s}, \mathbf{a}^{(k+1)}))$$

**Monotonic policy improvement theory**, specifically the policy improvement objective from Trust Region Policy Optimization (Schulman et al., 2015), can now be applied to this Diffusion MDP. This gives us the following proximal objective for policy improvement:

$$\eta(\mathbf{s}, \pi) \geq J_{\pi_{\text{ref}}}(\mathbf{s}, \pi) - \frac{4\epsilon\gamma}{(1-\gamma)^2}\left\{\max_{k, \mathbf{a}^{(k)}} D_{KL}(\pi_{\text{ref}}(\cdot \mid \mathbf{s}, \mathbf{a}^{(k)}), \pi(\cdot \mid \mathbf{s}, \mathbf{a}^{(k)}))\right\}$$
$$J_{\pi_{\text{ref}}}(\mathbf{s}, \pi) \triangleq \eta(\mathbf{s}, \pi_{\text{ref}}) + \mathbb{E}_{k \sim \text{Unif}\{1, K\}, \mathbf{a}^{(k)} \sim \pi_{\text{ref}} \atop \mathbf{a}^{(k-1)} \sim \pi(\cdot \mid \mathbf{a}^{(k)}, \mathbf{s})}[\gamma^k A_{\pi_{\text{ref}}}^{\text{diffusion}}((\mathbf{s}, \mathbf{a}^{(k)}), \mathbf{a}^{(k-1)})]$$

This result provides a lower bound on our policy objective equation 11, expressed through a loss that depends on the Diffusion MDP advantage of the reference policy $A_{\pi_{\text{ref}}}^{\text{diffusion}}$ and the KL divergence between the reference

policy $\pi_{\text{ref}}$ and our learning policy $\pi$. When our policy remains sufficiently close to the reference, maximizing this lower bound guarantees that performance will either improve or at least not deteriorate. We can therefore make progress by following the gradient of the advantage term in this lower bound:

$$\nabla_\phi J_{\pi_{\text{ref}}}(\mathbf{s}, \pi_\phi) = \nabla_\phi \underbrace{\mathbb{E}_{k \sim \text{Unif}\{1,K\}, \mathbf{a}^{(k)} \sim \pi_{\text{ref}}}\left[\gamma^k \mathbb{E}_{\mathbf{a}^{(k-1)} \sim \pi_\phi}[Q_{\pi_{\text{ref}}}^{\text{diffusion}}((\mathbf{s}, \mathbf{a}^{(k)}), \mathbf{a}^{(k-1)})]\right]}_{J_{\pi_{\text{ref}}}^{\text{proximal}}(\mathbf{s}, \pi_\phi)} \tag{12}$$

**Simplification via a One-Step Lower Bound:** Learning the multi-step value function $Q_{\pi_{\text{ref}}}^{\text{diffusion}}$ is still difficult. To make our objective practical, we introduce a key simplification based on a core property of diffusion models. We utilize the fact that the denoising function $\hat{\mathbf{a}} = D_\phi(\mathbf{a}^{(k)}, \sigma; \mathbf{s})$ aims to predict a strictly cleaner signal with each step (Song et al., 2020a). For a well-trained model, more sampling steps generally lead to higher quality actions $\hat{\mathbf{a}}$ (Ho et al., 2020; Song et al., 2020b). Since the $Q$-value is analogous to an unnormalized log probability (Levine, 2018), we build on the intuition that actions generated with more refinement steps should, on average, yield higher Q-values. This insight allows us to posit that the value of a single-step generation can serve as a tractable lower bound for the full, multi-step generation. We formalize this crucial simplification in the following proposition:

**Proposition 1.** *Define the data distribution under diffusion step $k \geq 0$ to be (with $\hat{\sigma} \geq \sigma_{min}$):*

$$p_k(\mathbf{a}|\mathbf{s}) = \int \pi(\mathbf{a}^{(0)}|\mathbf{s})\mathcal{N}(\mathbf{a}|\mathbf{a}^{(0)}, \sigma_k^2 \mathbf{I})\mathrm{d}\mathbf{a}^{(0)} \quad \hat{p}_k(\mathbf{a}|\mathbf{s}) = \int p_{k+1}(\mathbf{u}|\mathbf{s})\mathcal{N}(\mathbf{a}|D_\phi(\mathbf{u}, \sigma_{k+1}; \mathbf{s}), \hat{\sigma}^2 \mathbf{I})\mathrm{d}\mathbf{u}$$

*Under the assumptions that (i) entropy of $p_k$ and $\hat{p}_k$ strictly increases with $k$; (ii) the KL distance from $p_k$ and $\hat{p}_k$ to optimal policy $p^*$ is non-decreasing with $k$; (iii) $\hat{\sigma}$ is sufficiently large such that $D_{KL}(\hat{p}_0 \parallel p^*) \geq D_{KL}(p_0 \parallel p^*)^2$. Then for any state $\mathbf{s}$ and diffusion step $k$, we have*

$$\mathbb{E}_{p_0}[Q] \geq \mathbb{E}_{\hat{p}_k}[Q] \tag{13}$$

Proposition 1 formalizes a simple intuition: a single denoising step always improves upon a noisier action distribution, even if it does not reach the final, clean action distribution. This insight is powerful because it justifies replacing the multi-step expectation in equation 12 with a more tractable one-step lower bound. Specifically, the expectation over the remaining $k-1$ steps can be lower-bounded by the expectation over a single denoising step:

$$\gamma^k \mathbb{E}_{\mathbf{a}^{(k-1)} \sim \pi_\phi}[Q_{\pi_{\text{ref}}}^{\text{diffusion}}((\mathbf{s}, \mathbf{a}^{(k)}), \mathbf{a}^{(k-1)})]$$
$$= \gamma^{2k-K} \mathbb{E}_{\mathbf{a}^{(k-1)} \sim \pi_\phi} \underbrace{\mathbb{E}_{\mathbf{a}^{(k-2)} \cdots \mathbf{a}^{(0)} \sim \pi_{\text{ref}}}}_{\text{Additional } k-1 \text{ SDE steps}} [Q(\mathbf{s}, \mathbf{a}^{(0)})] \geq \gamma^{2k-K} \underbrace{\mathbb{E}_{\mathcal{N}(\hat{\mathbf{a}}|D_\phi(\mathbf{a}^{(k)}, \sigma_k; \mathbf{s}); \hat{\sigma}^2 \mathbf{I})}[Q(\mathbf{s}, \hat{\mathbf{a}})]}_{\text{One step generation}}$$

**Diffusion Value Gradients:** This one-step lower bound is much easier to optimize since it avoids learning separate value functions $Q_{\pi_{\text{ref}}}^{\text{diffusion}}$ in the diffusion MDP. Substituting it into equation 12 yields:

$$J_{\pi_{\text{ref}}}^{\text{proximal}}(\mathbf{s}, \pi_\phi) \geq \mathbb{E}_{k \sim \text{Unif}\{1,K\}, \mathbf{a}^{(k)} \sim \pi_{\text{ref}}, \boldsymbol{\epsilon} \sim \mathcal{N}(0,\mathbf{I})}[(\gamma^{2k}/\gamma^K) \cdot Q(\mathbf{s}, D_\phi(\mathbf{a}^{(k)}, \sigma_k; \mathbf{s}) + \hat{\sigma}\boldsymbol{\epsilon})] \tag{14}$$

This simplified objective can be expressed in a form similar to the $L_2$ loss commonly used in diffusion models (Kingma et al., 2021). Setting $\pi_{\text{ref}} = \pi_\phi$, we arrive at the final D2AC policy loss:

$$\nabla_\phi L_\pi^{\text{simple}}(\phi) = \mathbb{E}_{\substack{\mathbf{a} \sim \pi_\phi(\cdot|\mathbf{s}) \\ \boldsymbol{\epsilon} \sim \mathcal{N}(0,I), k \sim \text{Unif}\{1,K\} \\ \mathcal{N}(\hat{\mathbf{a}}|D_\phi(\mathbf{a}+\sigma_k\boldsymbol{\epsilon}, \sigma_k; \mathbf{s}), \hat{\sigma}^2 \mathbf{I})}} \nabla_\phi\left[\lambda(k) \|\hat{\mathbf{a}} + \nabla_{\hat{\mathbf{a}}}Q(\mathbf{s}, \hat{\mathbf{a}}) - D_\phi(\mathbf{a}+\sigma_k\boldsymbol{\epsilon}, \sigma_k; \mathbf{s})\|^2\right] \tag{15}$$

---

[2]While these are formal assumptions, they are grounded in the empirical behavior of well-trained diffusion models

The function $\lambda(k)$ is the weighting according to the noise schedule. $\lambda(k) = \gamma^{2k}/\gamma^K$ in our derivation; in practice we find that simply setting $\lambda(k) = 1$ works well. Conceptually, this loss teaches the denoising network $D_\phi$ to predict a target that has been pushed slightly up the gradient of the Q-function. In essence, each denoising step learns to do so in a direction that greedily improves the action according to the critic. Compared to diffusion policy gradients equation 10, this value-gradient approach avoids BPTT and provides a more stable training signal, whose quality is critically dependent on the rich information provided by our distributional critic.

# 5    Experiments

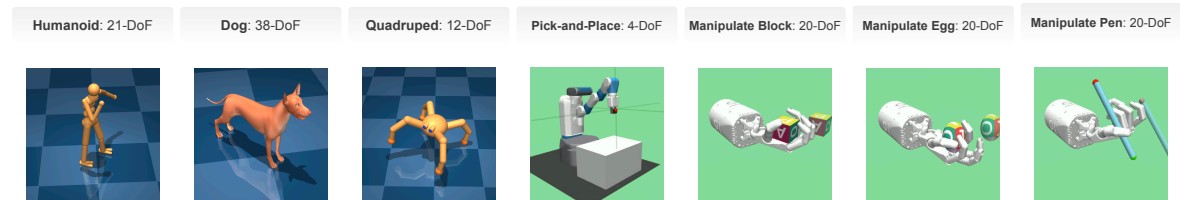

Figure 2: D2AC works out of the box across a wide range of environments, including locomotion and manipulation with sparse and dense rewards.

In these experiments, we evaluate the performance of D2AC across a range of domains to assess its generality, efficiency, and behavioral characteristics. We also investigate how D2AC compares to strong model-based baselines in selected tasks, aiming to understand how effectively it bridges the gap between model-free and model-based approaches.

**Evaluation environments.**    We evaluate D2AC across three primary domains: (i) the DeepMind Control Suite (Tassa et al., 2018), which consists mainly of locomotion tasks with dense rewards, (ii) the Multi-Goal RL environments from Plappert et al. (2018), which include robotic manipulation tasks with sparse rewards, and (iii) a predator–prey environment inspired by biological survival dynamics (Lai et al., 2024), which emphasizes adaptive behavior in dynamic and high-stakes scenarios. A list of the environments we use is shown in Figure 2. Those environments include 21-DoF Humanoid, 38-DoF Dog, 12-DoF Quadruped, a 4-DoF Pick-and-Place task, and a series of 20-DoF Shadow Hand tasks aiming to manipulate Block, Egg, and Pen to target orientations.

## 5.1    Benchmarking on dense-reward environments

In Figure 3, we showcase our benchmarking results on 12 tasks in the DeepMind control suite. We compare our method to a variety of baselines, including: Soft Actor Critic (Haarnoja et al., 2018), D4PG (Barth-Maron et al., 2018), and the model-free component of TD-MPC2 (Hansen et al., 2023), which we denote as *Two-hot + CDQ*, because it essentially combines SAC, two-hot critic representation, and clipped double Q-learning. We also compare against DIPO (Yang et al., 2023), which is a model-free RL method based on diffusion policy. D2AC achieves higher asymptotic performance across all 12 environments, and generally learns faster.

When compared to TD-MPC2 (Hansen et al., 2023) trained with the same number of environment interactions, we see that D2AC is very competitive despite being model-free. In Figure 4, we see that D2AC achieves performance within five percent of TD-MPC2 on 8 out of 12 benchmark environments. In 4 of the environments, the performance of D2AC exceeds TD-MPC2. The two standout environments were Dog Walk and Humanoid Stand, where D2AC significantly outpaced TD-MPC2 performance. This comparison is particularly revealing when considering the performance of the *Two-hot + CDQ* baseline, which represents the model-free component of TD-MPC2. Its relatively poor performance suggests that the strength of TD-MPC2 does not lie in its model-free foundation alone, but heavily relies on its sophisticated planning module. In contrast, D2AC closes the performance gap to this state-of-the-art model-based method without any planning. Our proposed model-free learning algorithm is orthogonal to any planning algorithm and can be used as a drop-in replacement in any model-based method, such as TD-MPC2.

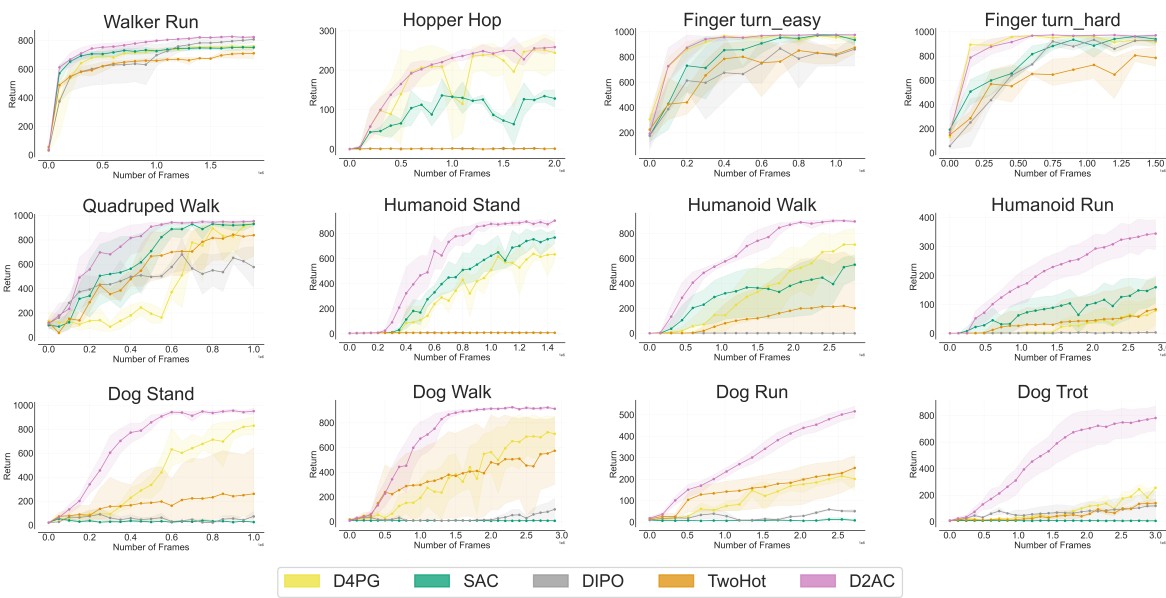

Figure 3: Experiments on **DeepMind Control Suite**. Results over 5 seeds. In model-free RL, D2AC achieves much better sample efficiency and asymptotic performance compared to all other baselines.

**D2AC is highly performant in a wide variety of settings:** D2AC achieves significantly better sample efficiency and asymptotic performance compared to all other model-free baselines we tested. This is true across multiple control modalities (grasping, locomotion, manipulation). Thanks to being model-free, **our method is considerably faster to run compared to TD-MPC2**: see Figure 10 for a clear comparison and see Appendix A for a detailed discussion. In particular, D2AC can make significant progress on the Dog Trot task within 24 hours, where TD-MPC2 has barely started increasing its episodic return; SAC completely fails on this task.

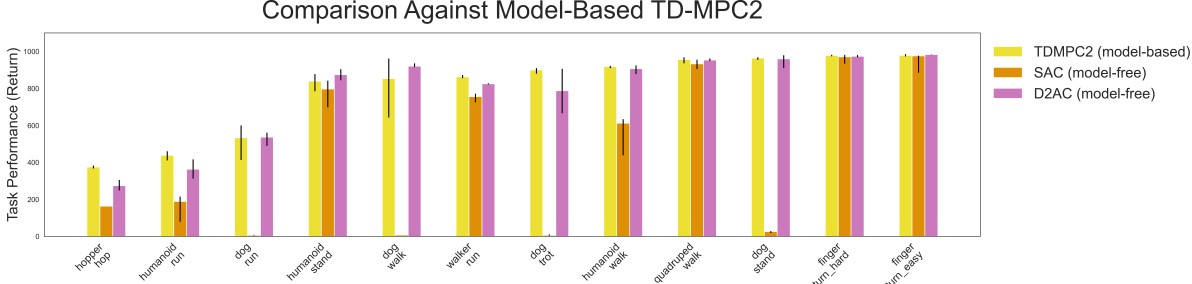

Figure 4: Comparison between model-based TD-MPC2 (Hansen et al., 2023), SAC (Haarnoja et al., 2018), and our method D2AC on DeepMind Control Suite. **D2AC without planning** can achieve results on-par with TD-MPC2.

Overall, these results challenge the notion that the recent success of model-based RL stems entirely from explicit planning with rollouts. The strong performance of D2AC suggests that the core principles often associated with model-based planning—specifically, the ability to model a distribution of possible returns and to iteratively refine action proposals-can be effectively distilled into a purely model-free framework.

## 5.2 Benchmarking on goal-conditioned RL environments

In Figure 5, we present results on robotic manipulation tasks with sparse rewards. All methods are trained with Hindsight Experience Replay (HER) (Andrychowicz et al., 2017). The baselines include: DDPG (Lillicrap

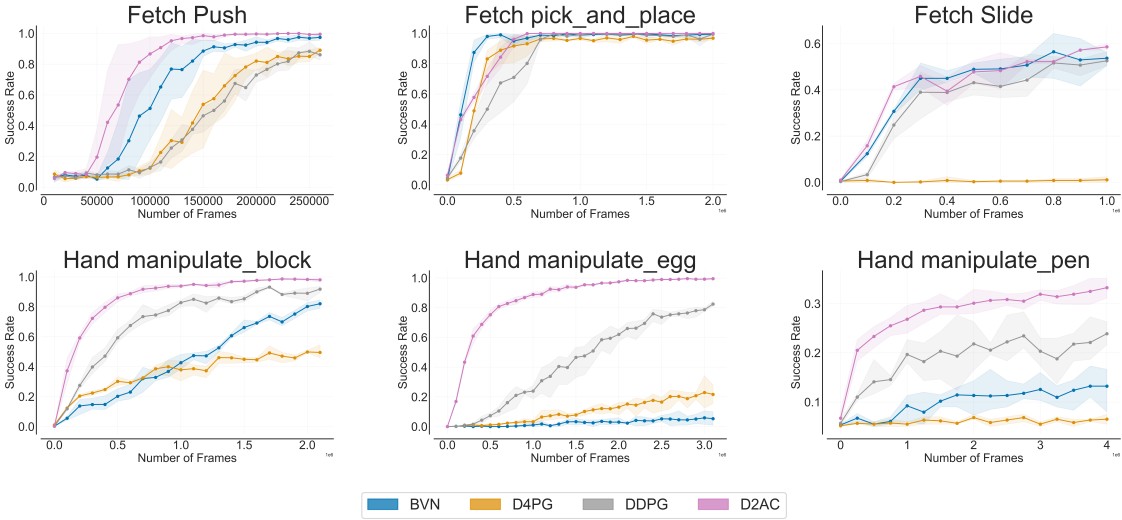

Figure 5: Experiments on **Multi-Goal RL environments with sparse rewards**. Results over 5 seeds.

et al., 2015) (the original algorithm used in the HER paper), a distributional version of HER (Eysenbach et al., 2019), and recently proposed Bilinear-Value Network (BVN) (Hong et al., 2022) found to be effective in goal-conditioned RL. We use a centralized learner and 20 sampling workers for all methods. We find that on Fetch tasks, D2AC performs better or on par with the previous SOTA method, BVN. On 20-DoF Hand Manipulate tasks, the second-best method is still DDPG, and our method can outperform it by a large margin, often both in terms of learning speed and policy performance at convergence. **This performance shows that D2AC is a strong base algorithm that can work out-of-the-box in a variety of settings.**

Interestingly, the previously proposed distributional critic function in Eysenbach et al. (2019) achieves worse performance than non-distributional critics. We hypothesize that this is because its specialized type of goal-conditioned distributional Bellman backup uses a one-hot target the moment a goal is reached, making it more difficult for the agent to learn how to consistently stay at the goal. Our distributional critics do not make any particular assumptions about the problem structure of goal-conditioned RL, but still achieve very strong results in this setting.

## 5.3 Biology-Inspired Benchmark: Behavioral Adaptation in Predator–Prey Scenarios

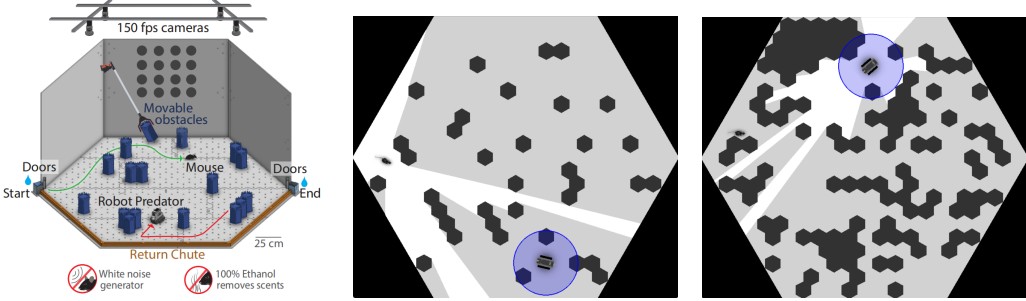

Figure 6: **Left:** Predator–prey arena(hexagonal maze, start to goal, +1 at goal, –1 when predator is within 0.1 units, matching the real-mouse water/air-puff setup). **Middle:** Prey's egocentric view on Map Level 5 (RL sim). **Right:** Predator's egocentric view on Map Level 9 (RL sim).

While standard reinforcement learning benchmarks primarily focus on task completion or reward maximization, they often overlook structural differences in the learned policies. To address this limitation, we introduce a predator–prey environment inspired by biological survival dynamics (Lai et al., 2024). Such scenarios serve as foundational models in both natural and artificial systems for analyzing adaptive decision-making

under pressure (Marrow et al., 1996; Han et al., 2025). This domain provides a natural testbed for analyzing the internal structure of learned policies, including diversity and strategic flexibility to adversarial changes, qualities often hidden behind reward curves.

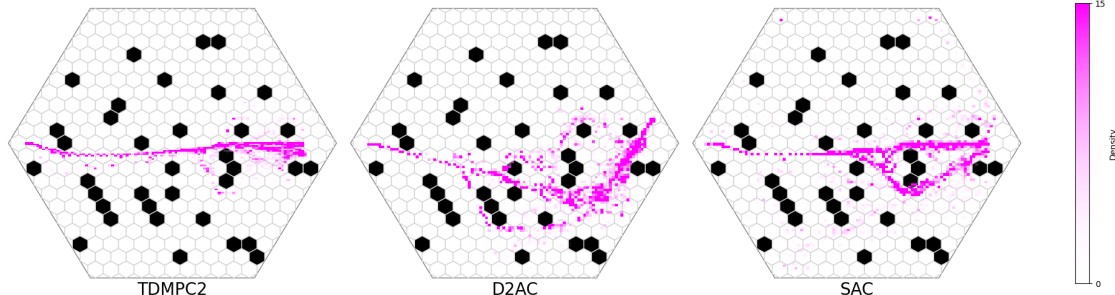

Figure 7: Illustrative example of state density for TD-MPC2, D2AC, and SAC in the predator-prey environment. Each visualization corresponds to 2000 sampled episodes from one representative run, where purple indicates the agent's state density as the prey reaches the goal while avoiding predators. Quantitative results in the main text are averaged over 5 independent runs.

Table 1: Exploration Coverage Analysis over five seeds (values reported as coverage $\%$ $\pm$ standard deviation)

| Algorithm | Visit Coverage $(\tau \geq 1)^{\mathrm{a}}$ | Main Coverage $(\tau \geq 10)^{\mathrm{b}}$ |
|---|---|---|
| SAC | $35.07 \pm 6.07$ | $10.18 \pm 1.18$ |
| TDMPC | $31.32 \pm 11.64$ | $10.09 \pm 1.76$ |
| D2AC | $\mathbf{44.58 \pm 3.16}$ | $\mathbf{20.04 \pm 0.49}$ |

*Notes.* [a] States visited at least once. [b] States visited at least 10 times.

**Metrics:** We train three agents (TD-MPC2, SAC, and our proposed D2AC) for 50,000 steps on Map Level 5 and evaluate: (i) *Exploration Coverage* (Table 1) through visit coverage and core-area coverage; (ii) *Survival Rate* over 2,000 episodes; (iii) *Zero-Shot Transfer* to unseen Map Level 9 (Table 2).

Table 2: Zero-shot transfer performance from Map Level 5 (trained) to Map Level 9 (unseen). Results are reported as survival rate $\%$ $\pm$ standard deviation over five seeds.

| Algorithm | Level 5 (Trained) | Level 9 (Zero-Shot) | **Change (%)** |
|---|---|---|---|
| TD-MPC2 | $72.33 \pm 1.95$ | $62.35 \pm 1.36$ | $-9.98$ |
| SAC | $86.37 \pm 3.23$ | $75.95 \pm 7.41$ | $-10.42$ |
| D2AC | $\mathbf{87.05 \pm 4.24}$ | $\mathbf{90.69 \pm 3.56}$ | $\mathbf{+3.64}$ |

**Results and Analysis:** As shown in Table 1, D2AC achieves the highest visit coverage (44.6%, 9.5% over SAC and 13.2% over TD-MPC2) and doubles TD-MPC2's core coverage, suggesting more comprehensive exploration. Interestingly, we observe a potential connection between exploration patterns and generalization: In zero-shot transfer (Table 2), both TD-MPC2 and SAC degrade by approximately 10%, while D2AC *improves* by 3.6%. Figure 7 further reveals that TD-MPC2 and SAC exhibit relatively consistent trajectories, while D2AC demonstrates more varied paths and appears to adapt more dynamically to predator movements. These findings suggest our framework not only achieves competitive survival rates but may also develop more flexible behavioral strategies.

**Insights on D2AC's Performance:** The results suggest D2AC's approach offers advantages in this environment. We hypothesize that the exploratory benefit emerges from our architecture—the distributional critic provides richer uncertainty information while the diffusion actor generates diverse action proposals through

iterative sampling. However, we acknowledge that higher exploration doesn't universally guarantee better performance—in some domains, focused exploitation might be preferable. The predator-prey environment specifically rewards strategic diversity, making it suitable for showcasing D2AC's exploration characteristics.

## 5.4 Distributional Critic vs. Diffusion Actor

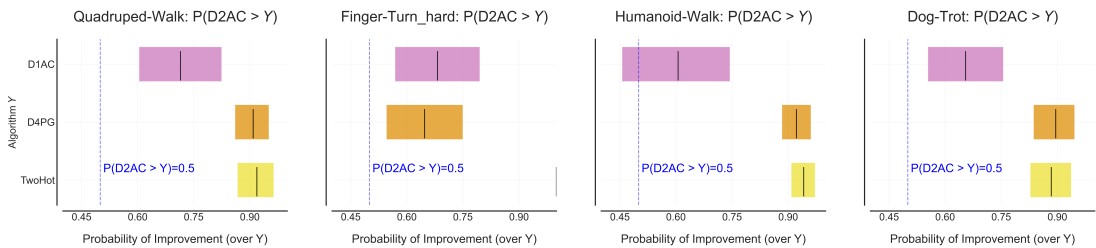

Figure 8: Ablation Studies based on Probability of improvement (Agarwal et al., 2021) using the recommended protocol from rliable. The X-axis gives the probability that D2AC improves upon the performance of the baseline algorithm on the Y-axis. The boxes represent 95% confidence intervals for each baseline.

First, we ask whether our diffusion actor is indeed critical. We test this by introducing D1AC, which uses our clipped double distributional critic but a standard Tanh-Gaussian actor. As shown in Figure 8, $P(\text{D2AC} > Y)$ is consistently above 0.5 for all choices of algorithm $Y$. In particular, the high probability of D2AC improving over D1AC (purple bars) directly demonstrates the benefit of the diffusion actor.

Beyond the actor, we also observe that D1AC remains competitive against strong distributional baselines such as D4PG and the model-free component of TD-MPC2 (Two-hot+CDQ). This seems to suggest that our proposed critic design provides substantial gains.

Hence, to directly isolate the contribution of our critic design, we conduct an ablation that disentangles its two key components: the distributional return modeling and the clipped double Q-learning mechanism. We evaluate four variants—our full D2AC, a version without clipped double Q-learning, a scalar critic with clipped double Q-learning, and a scalar critic with a single Q function—across three continuous control tasks.

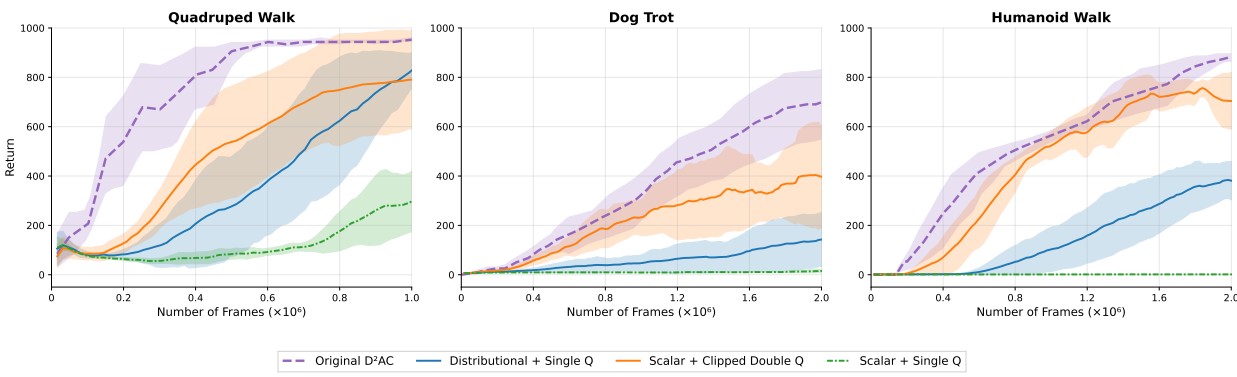

Figure 9: Ablation of critic components on three locomotion tasks. Both the distributional critic and clipped double Q-learning are essential, and the full D2AC achieves the best performance.

As shown in Figure 9, the full D2AC consistently outperforms all ablated versions. Removing either the distributional critic or the clipped double Q-learning leads to clear degradation, while eliminating both components results in the weakest performance. These results provide direct evidence that both mechanisms make additive and indispensable contributions to stability and overall performance.

## 5.5 Limitation of Our Method

Fundamentally, our method is still a model-free RL method; even a more expressive policy class based on denoising score matching cannot fully substitute for the role of planning. Although the denoising diffusion policy improvement we propose already has a lot of similarity to MPPI (Williams et al., 2015; 2016) in terms of optimization objective, MPPI is explicitly guided by the critic function during the sampling process, and simultaneously considers multiple possible trajectories. We expect that explicit value-based planning should be able to further enhance diffusion performance, but leave it for future work.

## 6 Conclusion

We have presented D2AC, a novel model-free RL algorithm that successfully integrates a powerful diffusion actor with a stable distributional critic. Our core technical contribution is a new, theoretically-grounded policy objective that makes training diffusion actors in an online setting both stable and efficient. This was enabled by our second contribution: a carefully designed distributional critic, stabilized with clipped double Q-learning, that provides the robust learning signal necessary for our actor to thrive. This synergistic design leads to a state-of-the-art method that excels on both sparse and dense reward tasks, runs significantly faster than model-based competitors, and demonstrates superior exploration and generalization on a challenging biology-inspired benchmark.

There exist several exciting avenues for future work. D2AC can be plugged into any model-based methods as the model-free RL component; we aim to explore how simple planning modules can boost its performance. While we built upon the C51 framework for our distributional critic, recent approaches like classification-based value learning (Farebrother et al., 2024) and histogram loss (Imani et al., 2024) offer promising alternatives that could potentially enhance D2AC's performance further. Finally, we are interested in improving the computational efficiency of D2AC by investigating faster diffusion models (Song et al., 2023; Song & Dhariwal, 2023; Chadebec et al., 2025) and beyond.

## Acknowledgments

We are grateful to Duruo Li (Department of Statistics, Northwestern University) for her valuable assistance with carefully verifying the correction of Proposition 1.

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

## A    Analysis of Computational Cost

To analyze the computational cost of D²AC, we compare its wall-clock training time against both the model-based TD-MPC2 and the model-free SAC (Figure 10). The results highlight D²AC's efficiency profile and its relationship with sample efficiency, as established in our main results (Figure 3).

**Computational Profile vs. Baselines:**   Our analysis reveals two distinct computational characteristics. First, as a model-free method, D²AC is dramatically more computationally efficient than the model-based TD-MPC2. It achieves superior or comparable performance in a fraction of the wall-clock time by avoiding the significant overhead of world model learning and online planning.

Second, when compared to the simpler model-free SAC, a clear trade-off emerges. SAC's lightweight architecture gives it a speed advantage in wall-clock time on less complex tasks. This is an expected consequence of D²AC's more powerful components, the diffusion actor and the distributional critic, which inherently require more computation per training step.

**Justifying the Computational Overhead with Sample Efficiency:**   The crucial question is whether this additional computational cost is justified. The answer lies in the significant gains in sample efficiency that D²AC achieves. As shown in our primary results (Figure 3), D²AC consistently extracts more information per environment interaction, leading to higher asymptotic performance.

This trade-off is most evident on challenging tasks like Humanoid Walk. While SAC is faster per iteration, its poor sample efficiency prevents it from solving the task. D²AC's computational investment per step, in contrast, enables a level of sample efficiency that is sufficient to master the task. Therefore, D²AC's design makes a highly effective trade-off: it modestly increases per-sample computation in exchange for the capability to solve complex problems that are intractable for simpler, faster methods.

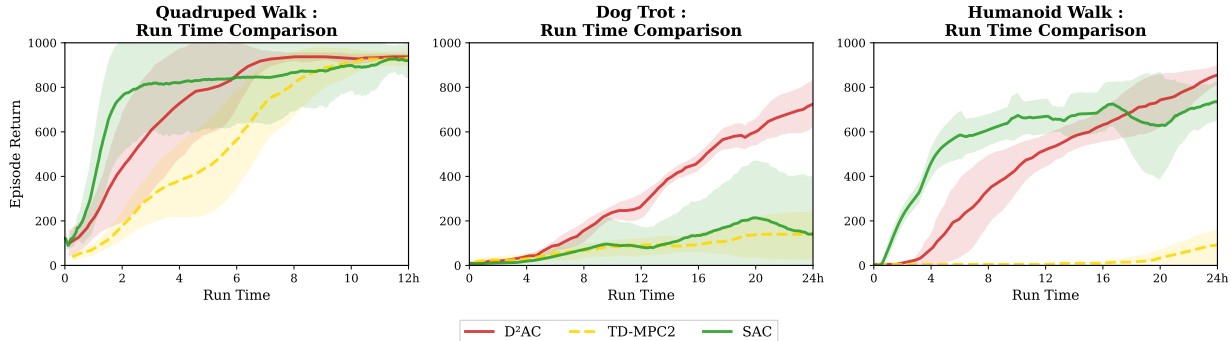

Figure 10: Wall-clock runtime comparison against TD-MPC2 and SAC on a single GPU. The figure highlights the differing computational profiles across tasks of varying complexity. While D²AC is substantially faster than the model-based TD-MPC2, its initial learning speed is surpassed by the simpler SAC on some tasks, illustrating a trade-off between computational overhead and the capacity for sustained learning on complex problems.

## B  Number of Diffusion Steps at Training vs the Final Model Performance

With respect to the diffusion actor, we are also interested in understanding the relationship between the number of diffusion steps at training vs the final model performance. In particular, we can consider both $K^{train}$, the number of diffusion steps used during training, and $K$, the number of diffusion steps used at inference. In Table 1, we see that there are some moderate improvements to the final results when training on longer diffusion horizons. Our results agree with the common practice in diffusion models (Song et al., 2020a; Karras et al., 2022) of using more diffusion steps for training and for inference. In addition, we also find that setting $\pi_{\text{ref}} = \pi_\phi$ as done in D2AC policy loss equation 15 rather than just using the actions from the replay buffer $\pi_{\text{ref}} \neq \pi_\phi$ slightly improves results.

Table 3: Performance Statistics at 500K Environment Steps

| Configuration | Quadruped Walk | Humanoid Walk |
|---|---|---|
| $K^{train} = K = 2$ | $778.6 \pm 227.6$ | $378.7 \pm 52.5$ |
| $K^{train} = K = 5$ | $\mathbf{941.5 \pm 9.7}$ | $\mathbf{425.1 \pm 39.9}$ |
| $K^{train} = 5, K = 2$ | $928.8 \pm 15.9$ | $386.9 \pm 59.8$ |
| $K^{train} = K = 5$ where $\pi_{\text{ref}} \neq \pi_\phi$ | $829.9 \pm 127.5$ | $314.2 \pm 76.5$ |

## C  On Two-hot Functions

The function $h_{\mathbf{z}_q}$ is called the *two-hot* function. Mathematically, $h_{\mathbf{z}_q}(z)$ is defined as:

$$h_{\mathbf{z}_q}(z)[k] = \begin{cases} 1 & z \leq V_{\min} \text{ and } k = 0, \\ 0 & z \leq \mathbf{z}_q[k-1], \\ \frac{z - \mathbf{z}_q[k-1]}{\mathbf{z}_q[k] - \mathbf{z}_q[k-1]} & \mathbf{z}_q[k-1] \leq z \leq \mathbf{z}_q[k], \\ \frac{\mathbf{z}_q[k+1] - z}{\mathbf{z}_q[k+1] - \mathbf{z}_q[k]} & \mathbf{z}_q[k] \leq z \leq \mathbf{z}_q[k+1], \\ 0 & z \geq \mathbf{z}_q[k+1], \\ 1 & z \geq V_{\max} \text{ and } k = N. \end{cases}$$

Which is a piece-wise linear function with the following properties. $\forall z$, if $V_{\min} \leq z \leq V_{\max}$,

$$\sum_{j=0}^{N} h_{\mathbf{z}_q}(z)[j] = 1 \quad \sum_{j=0}^{N} h_{\mathbf{z}_q}(z)[j] \cdot \mathbf{z}_q[j] = z \tag{16}$$

Which means that for any $z$ value, the two-hot function will distribute probability mass in the $[V_{\min}, V_{\max}]$ region that sums up to 1, with a weighted mean equal to $z$ itself. Note that the two-hot function is not the only function that satisfies those properties; for instance, the HL-Gauss parameterization proposed in (Imani & White, 2018) can also satisfy those two conditions.

Lately, a different type of discrete critic based on *two-hot* representation has gained popularity from the success of MuZero (Schrittwieser et al., 2020), Dreamer-v3 (Hafner et al., 2023), and TD-MPC2 (Hansen et al., 2023). To draw the connection between the two types of discrete critics, we first notice the following:

$$\sum_{j=0}^{N} \mathbf{p}[j] = 1 \qquad r + \gamma \sum_{j=0}^{N} \mathbf{z}_q[j] \cdot \mathbf{p}[j] = \sum_{j=0}^{N} (r + \gamma \mathbf{z}_q)[j] \cdot \mathbf{p}[j]$$

It follows that: two-hot discrete critic learning can be reformulated as **simply swapping the order between the sum and the two-hot function in the projection operator of distributional RL**:

$$\mathbf{\Phi}_{\mathrm{dist}}(\mathbf{z}_p, \mathbf{p}, \mathbf{z}_q)[k] = \sum_{j=0}^{N} h_{\mathbf{z}_q}(\mathbf{z}_p[j])[k] \cdot \mathbf{p}[j]$$

$$\mathbf{\Phi}_{\mathrm{twohot}}(\mathbf{z}_p, \mathbf{p}, \mathbf{z}_q)[k] = h_{\mathbf{z}_q}\left(\sum_{j=0}^{N} \mathbf{z}_p[j] \cdot \mathbf{p}[j]\right)[k]$$

And the two-hot label is:

$$\mathbf{a}' \sim \pi_\phi(\cdot \mid \mathbf{s}') \quad \boldsymbol{\ell}_{\mathrm{twohot}} = \mathbf{\Phi}_{\mathrm{twohot}}(r + \gamma \mathbf{z}_q, \mathbf{q}_{\bar{\theta}}(\mathbf{s}', \mathbf{a}'), \mathbf{z}_q)$$

$$L(\theta) = -\boldsymbol{\ell}_{\mathrm{twohot}}^{\top} \log \mathbf{q}_\theta(\mathbf{s}, \mathbf{a})$$

In comparison, distributional RL can be seen as using soft labels rather than hard labels; it allows the uncertainty about future returns to propagate through Q-learning. Two-hot discrete critic learning does not allow such uncertainty to propagate between different timesteps through Bellman backup.

## D  Policy Gradients for Variance-Exploding (VE) Diffusion

Let one step of the diffusion process $\mathbf{a}^{(k)} \to \mathbf{a}^{(k-1)}$ for policy $\pi_\phi$ be:

$$\pi_\phi(\mathbf{a}^{(k-1)} \mid \mathbf{a}^{(k)}, \mathbf{s}) = \mathcal{N}(\mathbf{a}^{(k-1)} \mid \boldsymbol{\mu}_\phi(\mathbf{a}^{(k)}, k; \mathbf{s}), (\sigma_k^2 - \sigma_{k-1}^2)\mathbf{I})$$

$$\boldsymbol{\mu}_\phi(\mathbf{a}^{(k)}, k; \mathbf{s}) = \mathbf{a}^{(k)} + (\sigma_k^2 - \sigma_{k-1}^2)\nabla_{\mathbf{a}} \log p(\mathbf{a}^{(k)} \mid \sigma_k, \mathbf{s})$$

$$\nabla_{\mathbf{a}} \log p(\mathbf{a} \mid \sigma, \mathbf{s}) = (D_\phi(\mathbf{a}, \sigma; \mathbf{s}) - \mathbf{a})/\sigma^2$$

where we can see that

$$\boldsymbol{\mu}_\phi(\mathbf{a}^{(k)}, k; \mathbf{s}) = \mathbf{a}^{(k)} + \frac{(\sigma_k^2 - \sigma_{k-1}^2)}{\sigma_k^2}\left(D_\phi(\mathbf{a}^{(k)}, \sigma_k; \mathbf{s}) - \mathbf{a}^{(k)}\right)$$

$$= \frac{\sigma_{k-1}^2}{\sigma_k^2}\mathbf{a}^{(k)} + \frac{(\sigma_k^2 - \sigma_{k-1}^2)}{\sigma_k^2}D_\phi(\mathbf{a}^{(k)}, \sigma_k; \mathbf{s})$$

In addition, let the latent variables $\mathbf{a}^{(1)} \cdots \mathbf{a}^{(K)}$ be:

$$q(\mathbf{a}^{(k)} \mid \mathbf{a}) = \mathcal{N}(\mathbf{a}^{(k)} \mid \mathbf{a}, \sigma_k^2 \mathbf{I})$$

Then we have the marginal distribution denoted as:

$$p(\mathbf{a}^{(k)}) = \int p_{\mathrm{data}}(\mathbf{a})q(\mathbf{a}^{(k)} \mid \mathbf{a})\mathrm{d}\mathbf{a}$$

Under the assumption that $\sigma_{\max}$ is sufficiently large, the initial noise distribution is pre-defined as:

$$p(\mathbf{x}^{(K)}) = \mathcal{N}(\mathbf{0}, \sigma_{\max}^2 \mathbf{I})$$

The full policy distribution is:

$$\pi_\phi(\mathbf{a} \mid \mathbf{s}) = \int p(\mathbf{a}^{(K)}) \prod_{k=1}^K \pi_\phi(\mathbf{a}^{(k-1)} \mid \mathbf{s}, \mathbf{a}^{(k)}) \mathrm{d}\mathbf{a}_{1:K}$$

The distribution of $\mathbf{a}^{(k)}$ given $\mathbf{a}^{(i)}$, for $k > i$, is given by:

$$q(\mathbf{a}^{(k)} \mid \mathbf{a}^{(i)}) = \mathcal{N}(\cdot \mid \mathbf{a}^{(i)}, (\sigma_k^2 - \sigma_i^2)\mathbf{I})$$

Since the forward process is Markov, we have

$$q(\mathbf{a}^{(k)}, \mathbf{a}^{(i)} \mid \mathbf{a}) = q(\mathbf{a}^{(k)} \mid \mathbf{a}) \cdot q(\mathbf{a}^{(i)} \mid \mathbf{a})$$

The posterior distribution can be written as:

$$q(\mathbf{a}^{(i)} \mid \mathbf{a}^{(k)}, \mathbf{a})$$

Which can be computed in closed form using the conjugate prior of Gaussian distributions:

$$q(\mathbf{a}^{(i)} \mid \mathbf{a}^{(k)}, \mathbf{a}) = \mathcal{N}\left( \cdot \left| \frac{\sigma_i^2}{\sigma_k^2}\mathbf{a}^{(k)} + \frac{(\sigma_k^2 - \sigma_i^2)}{\sigma_k^2}\mathbf{a}, \sigma_i^2 \frac{(\sigma_k^2 - \sigma_i^2)}{\sigma_k^2} \right. \right)$$

Then Jensen's inequality gives us the following lower bound (Sohl-Dickstein et al., 2015; Ho et al., 2020):

$$-\log \pi_\phi(\mathbf{a} \mid \mathbf{s}) \leq L_{\mathrm{prior}} + \sum_{k=1}^K \mathbb{E}_{q(\mathbf{a}^{(k)}|\mathbf{a})} D_{KL}\left[ q(\mathbf{a}^{(k-1)}|\mathbf{a}^{(k)}, \mathbf{a}) \,\|\, \pi_\phi(\mathbf{a}^{(k-1)}|\mathbf{a}^{(k)}, \mathbf{s}) \right]$$

$$L_{\mathrm{prior}} = D_{KL}(q(\mathbf{a}^{(K)} \mid \mathbf{a}) \,\|\, p(\mathbf{a}^{(K)}))$$

From the result of (Kingma et al., 2021), we know that

$$D_{KL}(q(\mathbf{a}^{(k-1)}|\mathbf{a}^{(k)}, \mathbf{a}) \,\|\, \pi_\phi(\mathbf{a}^{(k-1)}|\mathbf{a}^{(k)}, \mathbf{s})) = \frac{1}{2\sigma_{k-1}^2} \frac{\sigma_k^2 - \sigma_{k-1}^2}{\sigma_k^2} \|\mathbf{a} - D_\phi(\mathbf{a}^{(k)}, \sigma_k; \mathbf{s})\|^2$$

$$= \frac{1}{2}\left( \frac{1}{\sigma_{k-1}^2} - \frac{1}{\sigma_k^2} \right) \|\mathbf{a} - D_\phi(\mathbf{a}^{(k)}, \sigma_k; \mathbf{s})\|^2$$

Allowing us to arrive at the following result:

$$\log \pi_\phi(\mathbf{a}|\mathbf{s}) \exp Q(\mathbf{s}, \mathbf{a}) \geq -\mathbb{E}_{\boldsymbol{\epsilon} \sim \mathcal{N}(0,\mathbf{I}), k \sim \mathrm{Unif}(1 \cdots K)} [\lambda_{\mathrm{pg}}(k)\|\mathbf{a} - D_\phi(\mathbf{a} + \sigma_k\boldsymbol{\epsilon}, \sigma_k; \mathbf{s})\|^2 \exp Q(\mathbf{s}, \mathbf{a})]$$

where the weightings for policy gradient objective $\lambda_{\mathrm{pg}}$ are:

$$\lambda_{\mathrm{pg}}(k) = (1/\sigma_{k-1}^2 - 1/\sigma_k^2) \cdot (K/2)$$

## E  Proof of Proposition 1

**Proposition 1.** *Define the data distribution under diffusion step $k \geq 0$ to be (with $\hat{\sigma} \geq \sigma_{min}$):*

$$p_k(\mathbf{a}|\mathbf{s}) = \int \pi(\mathbf{a}^{(0)}|\mathbf{s})\mathcal{N}(\mathbf{a}|\mathbf{a}^{(0)}, \sigma_k^2\mathbf{I})\mathrm{d}\mathbf{a}^{(0)} \tag{17}$$

$$\hat{p}_k(\mathbf{a}|\mathbf{s}) = \int p_{k+1}(\mathbf{u}|\mathbf{s})\mathcal{N}(\mathbf{a}|D(\mathbf{u}, \sigma_{k+1}; \mathbf{s}), \hat{\sigma}^2\mathbf{I})\mathrm{d}\mathbf{u} \tag{18}$$

*Under the assumptions that (i) the entropy of $p_k$ and $\hat{p}_k$ strictly increases with $k$; (ii) the KL distance from $p_k$ and $\hat{p}_k$ to the optimal policy $p^*$ is **non-decreasing** with $k$; (iii) $\hat{\sigma}$ is sufficiently large such that $D_{\mathrm{KL}}(\hat{p}_0 \parallel p^*) \geq D_{\mathrm{KL}}(p_0 \parallel p^*)$. Then for any state $\mathbf{s}$ and diffusion step $k \geq 0$, we have*

$$\mathbb{E}_{p_0}[Q] \geq \mathbb{E}_{\hat{p}_k}[Q]$$

*Proof.* Fix an arbitrary state $s \in \mathcal{S}$. Because the Boltzmann optimal policy satisfies

$$p^*(a \mid s) = \frac{\exp\{Q(s,a)\}}{Z(s)}, \qquad Z(s) = \int \exp\{Q(s,a')\}\, da',$$

It follows that $Q(s,a) = \log Z(s) - \log p^*(a \mid s)$. For any distribution $q(\cdot \mid s)$ this identity yields

$$\mathbb{E}_q[Q] = \log Z(s) - D_{\mathrm{KL}}\big(q \parallel p^*\big) - \mathcal{H}(q), \tag{19}$$

where $\mathcal{H}(\cdot)$ denotes differential entropy. Applying equation 19 to $q = p_0$ and $q = \hat{p}_k$ and taking their difference gives

$$\mathbb{E}_{p_0}[Q] - \mathbb{E}_{\hat{p}_k}[Q] = \underbrace{D_{\mathrm{KL}}\big(\hat{p}_k \parallel p^*\big) - D_{\mathrm{KL}}\big(p_0 \parallel p^*\big)}_{:=\Delta_{\mathrm{KL}}(k)} + \underbrace{\mathcal{H}(\hat{p}_k) - \mathcal{H}(p_0)}_{:=\Delta_H(k)}. \tag{20}$$

**KL term.** Assumption (ii) states that the function $k \mapsto D_{\mathrm{KL}}\big(\hat{p}_k \| p^*\big)$ is non-decreasing. Thus, for any $k \geq 0$, $D_{\mathrm{KL}}\big(\hat{p}_k \| p^*\big) \geq D_{\mathrm{KL}}\big(\hat{p}_0 \| p^*\big)$. Assumption (iii) provides that $D_{\mathrm{KL}}\big(\hat{p}_0 \| p^*\big) \geq D_{\mathrm{KL}}\big(p_0 \| p^*\big)$. Combining these inequalities yields the following chain:

$$D_{\mathrm{KL}}\big(\hat{p}_k \| p^*\big) \geq D_{\mathrm{KL}}\big(\hat{p}_0 \| p^*\big) \geq D_{\mathrm{KL}}\big(p_0 \| p^*\big),$$

which implies that $\Delta_{\mathrm{KL}}(k) \geq 0$ for all $k \geq 0$.

**Entropy term.** Assumption (i) states that entropy is strictly increasing with $k$, which ensures $\mathcal{H}(\hat{p}_k) \geq \mathcal{H}(\hat{p}_0)$. The analysis thus reduces to showing $\mathcal{H}(\hat{p}_0) \geq \mathcal{H}(p_0)$.

Let $A^{(0)} \sim \pi(\cdot \mid s)$, and let $\varepsilon_0 \sim \mathcal{N}(0, \sigma_1^2 I)$ and $\varepsilon_1 \sim \mathcal{N}(0, \hat{\sigma}^2 I)$ be independent Gaussian noises. By definition, the distribution of $A^{(0)} + \varepsilon_0$ is $p_1$, and the distribution of $Y := D(A^{(0)} + \varepsilon_0, \sigma_1; s) + \varepsilon_1$ is $\hat{p}_0$.

For independent random vectors $X$ and $Z$, Cover (1999, Lemma 17.2.1) asserts that

$$\mathcal{H}(X + Z) \geq \mathcal{H}(X). \tag{21}$$

Applying equation 21 with $X = D(A^{(0)} + \varepsilon_0, \sigma_1; s)$ and $Z = \varepsilon_1$ gives

$$\mathcal{H}(\hat{p}_0) = \mathcal{H}(Y) \geq \mathcal{H}\big(D(A^{(0)} + \varepsilon_0, \sigma_1; s)\big).$$

Furthermore, since a deterministic mapping cannot increase entropy, and by Assumption (i), we have:

$$\mathcal{H}\big(D(A^{(0)} + \varepsilon_0, \sigma_1; s)\big) \geq \mathcal{H}(p_1) > \mathcal{H}(p_0).$$

Chaining these results together yields $\mathcal{H}(\hat{p}_0) > \mathcal{H}(p_0)$, which in turn implies that $\Delta_H(k) = \mathcal{H}(\hat{p}_k) - \mathcal{H}(p_0) \geq 0$ for all $k \geq 0$.

**Conclusion.** Since both terms on the right-hand side of equation 20 are non-negative, their sum is non-negative. Consequently,

$$\mathbb{E}_{p_0}[Q] - \mathbb{E}_{\hat{p}_k}[Q] \geq 0 \quad \implies \quad \mathbb{E}_{p_0}[Q] \geq \mathbb{E}_{\hat{p}_k}[Q], \qquad \forall k \geq 0.$$

This completes the proof. $\square$

## F  MPPI

Model Predictive Path Integral Control (MPPI) applies an iterative process of sampling and return-weighted refinement (Williams et al., 2015; 2016; Nagabandi et al., 2020; Hansen et al., 2022). Initial trajectories are sampled from a noise distribution:

$$\mathbf{x}_{i=0}^{k=0\cdots K-1} \sim \mathcal{N}(\cdot \mid \boldsymbol{\mu}_{i=0}, \boldsymbol{\sigma}_{i=0}^2)$$
$$\boldsymbol{\mu}_{i=0} = \mathbf{0}$$
$$\boldsymbol{\sigma}_{i=0}^2 = \sigma_{\max}^2 \mathbf{I}$$

At the $i$-th iteration, MPPI samples $K$ action sequences $\mathbf{x}_i^{k=0\cdots K-1}$ from

$$\mathcal{N}(\cdot \mid \boldsymbol{\mu}_i, \boldsymbol{\sigma}_i^2)$$

uses stochastic (virtual) rollouts to estimate the $K$ empirical returns of each action sequences $\{R\}_{k=0}^{K-1}$. It then updates the sampling distribution according to

$$\boldsymbol{\mu}_{i+1} = \left(\sum_{k=0}^{K-1} \exp(R_k/\lambda)\mathbf{x}_i^k\right) \Big/ \left(\sum_{k=0}^{K-1} \exp(R_k/\lambda)\right)$$
$$\boldsymbol{\sigma}_{i+1}^2 = \left(\sum_{k=0}^{K-1} \exp(R_k/\lambda)(\mathbf{x}_i^k - \boldsymbol{\mu}_{i+1})^2\right) \Big/ \left(\sum_{k=0}^{K-1} \exp(R_k/\lambda)\right)$$

MPPI uses a distribution of returns to update the action proposals to higher-return regions.

## G  On Tanh Squashing of Actions

The most commonly used action distribution for continuous control is a Gaussian distribution with tanh squashing (Haarnoja et al., 2018). The policy $\pi_\phi$ outputs the mean $\boldsymbol{\mu}_\phi$ and log sigma, which we exponentiate to get $\boldsymbol{\sigma}_\phi$.

$$\mathbf{u} \sim \mathcal{N}(\cdot \mid \boldsymbol{\mu}_\phi(\mathbf{s}), \boldsymbol{\sigma}_\phi^2(\mathbf{s})) \quad \mathbf{a} = \tanh(\mathbf{u}) \tag{22}$$

where $\mathbf{u}$ is sampled using the reparameterization trick (Kingma & Welling, 2013): $\mathbf{u} = \boldsymbol{\mu}_\phi + \boldsymbol{\epsilon} \odot \boldsymbol{\sigma}_\phi, \boldsymbol{\epsilon} \sim \mathcal{N}(0, \mathbf{I})$.

Applying the change of variable formula, The change of variable formula says

$$\log \pi(\mathbf{a} \mid \mathbf{s}) = \log \mathcal{N}(\mathbf{u} \mid \boldsymbol{\mu}_\phi, \boldsymbol{\sigma}_\phi^2) - \sum_i \log(1 - \tanh^2(\mathbf{u}_i))$$

And the second part can be written as:

$$\begin{aligned}
\log(1 - \tanh^2(\mathbf{u}))) &= 2\log \operatorname{sech} \mathbf{u} \\
&= 2\log 2 - 2\log(e^{\mathbf{u}} + e^{-\mathbf{u}}) \\
&= 2\log 2 - 2\log(e^{\mathbf{u}}(1 + e^{-2\mathbf{u}})) \\
&= 2\log 2 - 2\mathbf{u} - 2\log(1 + e^{-2\mathbf{u}}) \\
&= 2\log 2 - 2\mathbf{u} - 2\operatorname{softplus}(-2\mathbf{u}) \\
&= 2[\log 2 - \mathbf{u} - \operatorname{softplus}(-2\mathbf{u})]
\end{aligned}$$

Thus, we use an easy-to-compute and numerically stable form of the log likelihood of the Tanh Gaussian distribution:

$$f^{\mathrm{ent}}(\mathbf{u}, \boldsymbol{\mu}, \boldsymbol{\sigma}) = \log \mathcal{N}(\mathbf{u} \mid \boldsymbol{\mu}, \boldsymbol{\sigma}^2) - 2 \cdot \mathbf{1}^\top[\log 2 - \mathbf{u} - \operatorname{softplus}(-2\mathbf{u})] \tag{23}$$

Thus, we can use the squashing to enforce the action boundary, and apply entropy regularization (Haarnoja et al., 2018):

$$L(\alpha) = -\alpha[f_\phi^{\mathrm{ent}} - \lambda_{\mathrm{ent}}|\mathcal{A}|]$$

We apply to policy network similar to other model-free baselines.

---

**Algorithm 1** Policy Optimization (*under Tanh Action Squashing*)

---

**procedure** UPDATE($\phi, \alpha \mid \mathbf{s}, \mathbf{a}, \sigma, Q$)
    **Input:** policy parameters $\phi$, entropy coefficient $\alpha$.
    **Input:** state $\mathbf{s}$, continuous action $\mathbf{a} \in (-1, 1)$.
    **Input:** differentiable $Q(\mathbf{s}, \mathbf{a})$, noise level $\sigma$.
    $\boldsymbol{\epsilon} \sim \mathcal{N}(0, \mathbf{I}), \quad \tilde{\mathbf{u}} = \tanh^{-1}(\mathbf{a}) + \boldsymbol{\epsilon} \cdot \sigma$
    $\hat{\boldsymbol{\mu}}_\phi = \boldsymbol{\mu}_\phi(\mathbf{s}, \tilde{\mathbf{u}}, \sigma), \quad \hat{\boldsymbol{\sigma}}_\phi = \boldsymbol{\sigma}_\phi(\mathbf{s}, \tilde{\mathbf{u}}, \sigma)$
    $\mathbf{x}_\phi \sim \mathcal{N}(\cdot \mid \hat{\boldsymbol{\mu}}_\phi, \hat{\boldsymbol{\sigma}}_\phi^2)$
    $\mathbf{x}^{\text{target}} = \mathbf{x} + \nabla_{\mathbf{x}} Q(\mathbf{s}, \tanh(\mathbf{x}))\big|_{\mathbf{x}=\mathbf{x}_\phi}$           ▷ Eq equation 15
    $\hat{f}_\phi^{\text{ent}} = f^{\text{ent}}(\mathbf{x}_\phi, \hat{\boldsymbol{\mu}}_\phi, \hat{\boldsymbol{\sigma}}_\phi)$           ▷ Eq equation 23
    $\phi \leftarrow \text{SGD}(\phi, \nabla_\phi[\|\mathbf{x}_\phi - \mathbf{x}^{\text{target}}\|^2 + \alpha \hat{f}_\phi^{\text{ent}}])$
    $\alpha \leftarrow \text{SGD}(\alpha, \nabla_\alpha[-\alpha(\hat{f}_\phi^{\text{ent}} - \lambda_{\text{ent}}|\mathcal{A}|)])$           ▷ Update temperature
**end procedure**

---

---

**Algorithm 2** : **D2 Actor Critic**

---

**procedure** ACTORCRITIC($\mathcal{D}, \theta, \phi, \alpha$)
    **Input:** $(\mathbf{s}, \mathbf{a}, \mathbf{s}', r) \sim \mathcal{D}$
    **Input:** critic parameters $\theta$, policy parameters $\phi$.
    **Input:** entropy coefficient $\alpha$.
    $\mathbf{a}' \sim \pi_\phi(\cdot \mid \mathbf{s}')$           ▷ Diffusion Sampling
    Optimize $\theta$ with clipped double distributional RL equation **??**
    $i \sim \text{Unif}\{1 \cdots K^{\text{train}}\}, \quad j \sim \text{Unif}\{1 \cdots K\}$
    $\sigma_i \leftarrow \sigma(\frac{i-1}{K^{\text{train}}-1}), \quad \sigma_j \leftarrow \sigma(\frac{j-1}{K-1})$           ▷ Select noise levels in EDM
    Update $\phi, \alpha$ on $(\mathbf{s}, \mathbf{a}, \sigma_i)$ and $(\mathbf{s}', \mathbf{a}', \sigma_j)$.           ▷ Alg 1
    Target network update: $\bar{\theta} \leftarrow \tau\bar{\theta} + (1 - \tau)\theta$           ▷ Exponential moving average.
**end procedure**

---

## H   Policy Network Parameterization

Let $p_\sigma(\tilde{\mathbf{u}}) = \int p_{\text{data}}(\mathbf{u})\mathcal{N}(\tilde{\mathbf{u}} \mid \mathbf{u}, \sigma^2\mathbf{I})d\mathbf{u}$. The min $\sigma_{\min}$ and max $\sigma_{\max}$ of noise levels are selected such that $p_{\sigma_{\min}}(\tilde{\mathbf{u}}) \approx p_{\text{data}}(\mathbf{u})$ and $p_{\sigma_{\max}}(\tilde{\mathbf{u}}) \approx \mathcal{N}(\mathbf{0}, \sigma_{\max}^2\mathbf{I})$. We adopt the EDM denoiser parameterization (Karras et al., 2022):

$$\boldsymbol{\mu}_\phi(\mathbf{s}, \mathbf{u}, \sigma) = c_{\text{skip}}(\sigma)\mathbf{u}$$
$$+ c_{\text{out}}(\sigma)F_\phi(\mathbf{s}, c_{\text{in}}(\sigma)\mathbf{u}, c_{\text{noise}}(\sigma)) \tag{24}$$

With the specific functions being:

$$c_{\text{skip}}(\sigma) = \sigma_{\text{data}}^2/(\sigma^2 + \sigma_{\text{data}}^2)$$

$$c_{\text{out}}(\sigma) = \sigma \cdot \sigma_{\text{data}}/\sqrt{\sigma^2 + \sigma_{\text{data}}^2}$$

$$c_{\text{in}}(\sigma) = 1/\sqrt{\sigma^2 + \sigma_{\text{data}}^2}$$

$$c_{\text{noise}}(\sigma) = \log\sigma$$

This parameterization offers powerful inductive bias for an iterative denoising process that has been shown to work well. $\boldsymbol{\sigma}_\phi(\mathbf{s}, \mathbf{a}, \sigma)$ uses the same network $F_\phi$ except for the last linear layer. The conditioning of $c_{\text{noise}}(\sigma)$ is done via positional embedding (Vaswani et al., 2017).

A sequence of noise levels is used $\sigma_{\min} = \sigma_1 < \sigma_2 < \cdots < \sigma_M = \sigma_{\max}$. We follow EDM in setting $\sigma_i = \sigma(\frac{i-1}{M-1})$ for $i = \{1, \cdots, M\}$ and $\rho = 7$:

$$\sigma(\eta) = \left(\sigma_{\min}^{1/\rho} + \eta\left(\sigma_{\max}^{1/\rho} - \sigma_{\min}^{1/\rho}\right)\right)^\rho \tag{25}$$

## I  Biology-inspired Environment Experiment Details

The experimental environment was designed to examine strategic escape behaviors in mice by approximating predator–prey interactions within a hexagonal arena (Lai et al., 2024), similar but not identical to the setup described in Han et al. (2025). In our implementation, a custom robotic device acted as a predator-like pursuer, delivering brief air puffs as an aversive stimulus when the simulated mouse was "caught", while the mouse's task was to reach a designated goal location without interception.

The agent operated in a continuous two-dimensional action space, where each action specified target coordinates (x,y) within the arena. Each simulation step corresponded to 0.25 seconds of behavior in the real-world mouse task.

The state was represented by a ten-dimensional vector that included the simulated mouse's position and heading, the robot's position when it was visible, the distance to the goal, and a binary variable indicating whether the mouse had just received an air puff.

And we receive a reward of +1 for reaching the goal and -1 when being caught. At the start of each episode, the predator was initialized outside the prey's field of view at a randomly chosen position. This initialization remained stochastic even when a fixed random seed was used, ensuring variability in the predator's starting state across episodes.

For all baselines, we adopted their official implementations and default hyperparameters (Haarnoja et al., 2018; Hansen et al., 2023). Specifically, both SAC and our method D2AC utilized a standard episodic setting, while TD-MPC2 was configured using its original non-episodic formulation as presented in its peer-reviewed paper.

## J  Hyperparameters

**Hyperparameter Tuning.**  D2AC demonstrates considerable robustness to most hyperparameters. For the majority of our experiments, we adopted standard values from prior literature for common parameters such as learning rates and network architectures, without performing extensive, task-specific grid searches. The consistent use of a single core set of hyperparameters across a diverse set of domains highlights the general applicability and stability of our algorithm.

The most critical task-specific consideration is the value range $[V_{\min}, V_{\max}]$ for the distributional critic. These parameters define the support for the return distribution and must be set to appropriately cover the plausible range of cumulative discounted rewards for a given task family. As detailed in our hyperparameter table 4, we set a wider range for the high-reward DeepMind Control Suite tasks (Tassa et al., 2018), whereas a narrower range was used for the sparse-reward, goal-conditioned manipulation tasks (Plappert et al., 2018).

Beyond setting a reasonable support for the return distribution, D2AC does not require laborious, per-environment tuning, underscoring its potential as a general-purpose reinforcement learning algorithm.

Table 4: Hyperparameters for D2AC

| Parameters | Value |
|---|---|
| Batch size | 256 |
| Optimizer | AdamW (Loshchilov & Hutter, 2017) |
| Learning rate for policy | 0.001 |
| Learning rate for critic | 0.001 |
| Learning rate for temperature $\alpha$ | 0.0001 |
| $\alpha$ initialization | 0.2 |
| Weight Decay | 0.0001 |
| Number of hidden layers (all networks) | 2 |
| Number of hidden units per layer | 256 |
| Non-linearity | Layer Normalization (Ba et al., 2016) + ReLU |
| $\gamma$ | 0.99 |
| $\lambda_{\text{ent}}$ | 0.0 |
| Polyak for target network | 0.995 |
| Target network update interval | 1 for dense rewards and predator-prey |
| | 10 for sparse rewards |
| Ratio between env vs optimization steps | 1 for dense rewards and predator-prey |
| | 2 for sparse rewards |
| Initial random trajectories | 200 |
| Number of parallel workers | 4 for dense rewards and predator-prey |
| | 20 for sparse rewards |
| Update every number of steps in environment | same as Number of parallel workers |
| Replay buffer size | $10^6$ for dense rewards and predator-prey |
| | $2.5 \times 10^6$ for sparse rewards |
| $[V^{\min}, V^{\max}]$ | $[-1000, 1000]$ for DM control |
| | $[-200, 200]$ for predator-prey |
| | $[-50, 0]$ for Multi-Goal RL |
| Number of bins (size of the support $\mathbf{z}_q$) | 201 for DM control |
| | 201 for predator-prey |
| | 101 for Multi-Goal RL |
| $\sigma_{\min}$ | 0.05 |
| $\sigma_{\max}$ | 2.0 |
| $\sigma_{\text{data}}$ | 1.0 |
| $\rho$ | 7 |
| $M$ | 2 |
| $M^{\text{train}}$ | 5 |
| Noise-level conditioning | Position encoding on $(10^3 \log \sigma)/4$ |
| Embedding size for noise-level conditioning | 32 |

