# OpenReview forum: "D2 Actor Critic: Diffusion Actor Meets Distributional Critic"
_TMLR — Accepted by TMLR_

### Review · Reviewer_WCJ3 · 2025-07-20

**Summary Of Contributions:**

This paper combines the distributional critic and diffusion actor to enhance the actor critic algorithm. This distributional critic is also equipped with clipped double Q-learning. Extensive experiments are conducted, including the classical locomotion tasks with dense rewards, manipulation tasks with sparse rewards, as well as a biologically motivated environment.

**Audience:**

Yes

**Claims And Evidence:**

Yes

**Requested Changes:**

Please make changes regarding the weaknesses I found above. There are other comments on the changes.

$q_\theta(s, a)$ in Section 3.3 should be the probability of each bin, which is not clearly explained.

Since D4PG also uses a categorical distributional critic loss, the only difference in the algorithm shown in Figure 3 between D2AC and D4PG is the diffusion policy. Is that correct?

What is the relationship between a larger visit coverage rate and better performance? It seems that the methodology does not involve any explanation of the exploration. Be aware that a high exploration coverage ratio does not always imply superior performance.

It is challenging to parse the results from Figure 8. More detailed explanations are expected in the revised version.

**Strengths And Weaknesses:**

## Strengths
- The proposed method is technically sound, although it is not new.
- The writing is clear and easy to follow.
- The experiments are rigorous, including classical dense rewards, sparse rewards, and a new biologically motivated environment to investigate the exploration coverage, which looks interesting.

## Weaknesses
- The proposed method has limited novelty. When it comes to distributional policy gradient methods, it is very common to consider the distributional critic loss, such as [1,2,3]. Also, the double learning technique is also used in the implementation of [2] as far as I know. It is easy to expect the performance improvement given the success of the distributional RL. The categorical distributional loss is also used in D4PG. On the other hand, in the model-free RL in the robotics literature, oftentimes, the actor is parameterized by a diffusion policy for an actor trajectory planning by leveraging the powerful expressiveness of the diffusion model. Therefore, combining these two ideas is not new, making this paper lack technical contribution. For example, it seems that [1] also considers the combination of diffusion policy and distributional critic. Although it is an arXiv paper, it is still useful to have a discussion about the difference from it.

- The technical descriptions are inaccurate or even erroneous. In the third paragraph of Section 1, (Bellmare 2017) did not consider the distributional critic. The concept ‘distribution’ is used to characterize the return random variable instead of Q function, which is the expectation of the return. Still in this paragraph, it is thus incorrect to say ‘distributional Q estimation.’ In section 2, it is also not accurate to say ‘value distribution’. This error occurs many times throughout this paper. A lack of understanding of the mechanism behind distributional reinforcement learning (RL) is problematic in writing and explanation of the empirical results in a deep way.

- It is not clear what the proposed diffusion policy's contribution is. Although I do not think there is anything new in the distributional critic part, I am trying to find some technical contributions in this section of the proposed diffusion policy. It seems that the content before equation (12) is known. The proposed simplifications may be new, although it seems straightforward by only conducting a one-step diffusion process. I kindly suggest the authors highlight the real contribution or the difference in the diffusion policy compared with others, which is more helpful to help me and readers to understand this paper. A large concern here is that the combination of the two techniques in this paper seems separate. For example, in the diffusion policy section, the authors still use the language of Q function, instead of explaining the mechanism in the context of learning the whole return distribution. This also reduces the technical contribution as far as I can tell.


### Reference
[1] Liu, Tong, et al. "Distributional Soft Actor-Critic with Diffusion Policy." arXiv preprint arXiv:2507.01381 (2025).
[2] Duan, Jingliang, et al. "Distributional soft actor-critic: Off-policy reinforcement learning for addressing value estimation errors." IEEE transactions on neural networks and learning systems 33.11 (2021): 6584-6598.
[3] Ma, Xiaoteng, et al. "DSAC: Distributional Soft Actor-Critic for Risk-Sensitive Reinforcement Learning." Journal of Artificial Intelligence Research 83 (2025).

---

> ### Author Response · Authors · 2025-08-05
>
> **Q: (Weakness 1 and 3) The proposed method has limited novelty... combining these two ideas is not new... the combination of the two techniques in this paper seems separate...**
>
> ---
>
> Thank you for this thoughtful and detailed feedback. We genuinely appreciate the time you invested in providing such constructive criticism, and we acknowledge that our original manuscript did not sufficiently articulate the unique contributions of our approach. We address your main concerns by grouping Weaknesses 1 and 3 into two parts.
>
> On Our Primary Technical Contribution:
>
> D2AC's main contribution is a new, theoretically grounded policy improvement objective that makes online training of diffusion policies practical and stable (Eq. 15). While concurrent work, such as [1], also explores combining distributional critics with diffusion policies, our approaches differ fundamentally. They optimize a policy objective adapted for diffusion actors, whereas D2AC derives a new objective that directly aligns each denoising step with Q-function gradients. This distinction matters for practical training. Policy optimization for diffusion models faces challenges with high estimation variance [2] due to the complex, multi-step sampling process. In contrast, D2AC addresses this issue from first principles. Starting from monotonic policy improvement theory, we formulate a "Diffusion MDP" and derive a principled single-step simplification via a formal lower bound (Proposition 1). This enables us to supervise each denoising step directly with the critic's guidance, avoiding the variance issues present in standard policy optimization approaches [1,3].
>
> On Our Purpose-Driven Actor-Critic Integration:
>
> Our distributional critic serves a specific purpose: providing the stable guidance signal that our policy update requires. We recognized that the primary challenge for online diffusion policy training is updating stability rather than maximizing information utilization. Therefore, we used a critic that combines well-established Clipped Double Q-learning with categorical distributions to deliver reliable Q-values. While these individual techniques are well-established, their combination specifically addresses the stability requirements of our new diffusion policy update. The integration is functional: our stable critic enables our actor update to work effectively. This approach achieves model-free performance comparable to model-based methods while being computationally faster. Importantly, our method works out-of-the-box across diverse domains, from dense-reward continuous control to sparse-reward goal-conditioned manipulation with HER, demonstrating broader applicability than concurrent work [1]. We also recognize that this conclusion requires more rigorous validation, so our ablation studies (D1AC vs D2AC) and the additional critic ablation experiments we just added confirm that this functional dependency is important for performance.
>
> We have substantially revised our Introduction, Related Works, Ablation Study, and Methodology sections to make these contributions and their relationships more explicit.
>
> ---
>
> [1] Liu, Tong, et al. "Distributional Soft Actor-Critic with Diffusion Policy." arXiv preprint arXiv:2507.01381 (2025).
>
> [2]Schulman, John, et al. "High-dimensional continuous control using generalized advantage estimation." arXiv preprint arXiv:1506.02438 (2015).
>
> [3] Ma, Haitong, et al. "Efficient Online Reinforcement Learning for Diffusion Policy." arXiv preprint arXiv:2502.00361 (2025).

---

> ### Author Response · Authors · 2025-08-29
>
> **Q: Since D4PG also uses a categorical distributional critic loss, the only difference in the algorithm shown in Figure 3 between D2AC and D4PG is the diffusion policy. Is that correct?**
>
> ----
>
> A: You are correct that the primary algorithmic difference is our diffusion actor. We should also clarify that there is another distinction: our critic integrates Clipped Double Q-learning with the categorical distribution, which is not present in the original D4PG.
>
> ----
>
> **Q: What is the relationship between a larger visit coverage rate and better performance? It seems that the methodology does not involve any explanation of the exploration...**
>
> ----
>
> A: We appreciate this important question, which has helped us realize we need to be more careful about our claims regarding exploration.
>
> First, we agree that high coverage does not always mean better performance. In our specific sparse-reward, adversarial predator-prey benchmark, thorough exploration is a prerequisite for finding robust policies. Our results support this: D2AC's higher coverage correlates with both better in-distribution performance and superior zero-shot transfer capabilities.
>
> Second, you are correct that we do not have an explicit exploration explanation. We only have a hypothesis: The exploratory advantage appears to emerge from our architecture. Our distributional critic models the entire return distribution, providing richer uncertainty information that helps the agent avoid risky shortcuts. Our diffusion actor generates diverse action proposals through its iterative sampling process, naturally encouraging broader exploration.
>
> We acknowledge this is an interesting empirical finding, and we have added this explanation to clarify our observations.
>
> ----
>
> **Q: Issues with technical terminology and unclear explanations (regarding distributional RL concepts, Section 3.3 notation, and Figure 8 analysis)**
>
> ----
>
> A: Thank you for pointing out these inaccuracies and unclear explanations. We take responsibility for the imprecision in our terminology. Following your guidance, we have revised the entire manuscript to correct instances of inaccurate terminology (e.g : changing value distribution to return distribution) and corrected our description of Bellemare et al. (2017). We have also clarified the notation in Section 3.3.
>
> We have also rewritten the analysis paragraph for Figure 8 in Section 5.4 to provide a clearer interpretation. Additionally, we provide a more targeted ablation study that conclusively demonstrates the additive contributions of both distributional return and double clipped Q-learning.

---

> > ### Comment · Reviewer_WCJ3 · 2025-09-03
> > **Further Comment**
> >
> > Thank you for the authors' rebuttal, which is helpful for me to further understand this paper. It is great for a revision in writing and highlights the contribution. I agree the distributional critic should be useful to stabilize the variance in the multi-step planning of the diffusion policy. However, the algorithmic novelty compared with [1] and D4PG is still very limited, as the change is straightforward. Beyond that, the authors are expected to put more effort into demonstrating the stability benefit of using distributional critic in diffusion policy. However, after I double-checked Proposition 1, I think it is weak for this demonstration. In particular, the monotonic decreasing of KL divergence or the increasing of entropy cannot be guaranteed in the real distribution critic algorithm. Given the reasons above, I suggest a substantial revision and algorithmic novelty in the next version of the paper in the future.

---

### Review · Reviewer_EjNa · 2025-08-16

**Summary Of Contributions:**

The paper introduces a new model-free actor-critic method that combines a distributional critic with clipped double Q learning and a parameterized actor based on a diffusion policy.
The paper also derives some lower bounds on their policy objective and introduces some simplifications to the diffusion policy to reduce the sampling steps required.
The paper evaluates the proposed algorithm across a wide range of benchmarks: DeepMind Control Suite, a biologically inspired environment, and some robotics tasks.

**Audience:**

Yes

**Broader Impact Concerns:**

no concerns.

**Claims And Evidence:**

Yes

**Requested Changes:**

- I would like to see more discussion on the computational complexity of D2AC with a runtime comparison to other model-free algorithms such as SAC.
- I didn't fully understand the lower bound discussion Eq. 13/Eq.14, I think clarifying these parts more with more discussion on how to interpret this lower bound would make the paper better.

**Strengths And Weaknesses:**

### Strengths:
- The paper is mostly well-written, and the contributions are clear.
- Combining distributional RL with diffusion policy is a novel and promising idea.
- The paper extensively evaluated their algorithm across several domains and against several baselines, including model-based RL methods.
- The proposed algorithm shows strong performance across several domains explored in the paper.
### Weaknesses:
-  I am still unclear about the computational complexity of the proposed algorithm. I know there are some run-time experiments in Figure 9, but it was only comparing with model-based, not model-free RL.
- There are different ways and losses to learn a distributional critic that has been used for RL before, such as in [1,2]; the paper lacks discussion on those other options.

[1] Imani, Ehsan, et al. "Investigating the histogram loss in regression." arXiv preprint arXiv:2402.13425 (2024).
[2] Farebrother, Jesse, et al. "Stop regressing: Training value functions via classification for scalable deep rl." arXiv preprint arXiv:2403.03950 (2024).

---

> ### Author Response · Authors · 2025-08-29
>
> **Q: I am still unclear about the computational complexity of the proposed algorithm. I know there are some run-time experiments in Figure 9, but it was only comparing with model-based, not model-free RL.**
>
> ---
>
> We thank the reviewer for this crucial question regarding computational cost. You are correct that our initial comparison focused on model-based methods. To provide a more complete picture, we have added Appendix A: Analysis of Computational Cost, which now includes a direct wall-clock runtime comparison between D2AC, SAC, and TD-MPC2.
>
> Our new analysis confirms that due to its simpler architecture, SAC is indeed faster than D2AC on a per-step basis, while both model-free methods are significantly faster to train than TD-MPC2. However, the analysis highlights that this raw speed does not translate to success on more challenging tasks. The core finding is that D2AC makes a deliberate and effective trade-off: it accepts a modest increase in computational overhead per step to achieve the superior sample efficiency required to solve complex problems where simpler methods like SAC fail. Therefore, this computational investment is justified by its ability to succeed on a much wider and more challenging range of tasks.
>
> ---
>
> **Q: I didn't fully understand the lower bound discussion Eq. 13/Eq.14, I think clarifying these parts more with more discussion on how to interpret this lower bound would make the paper better.**
>
> ---
>
> We appreciate your valuable feedback. We acknowledge that the discussion of the lower bound (Eq. 13/14) in the original manuscript lacked clarity, and we will elaborate on its interpretation and significance in the revised version.
>
> The core challenge is that directly optimizing the multi-step objective in Eq. 12 is computationally intractable due to the $Q^{diffusion}_{\pi \ ref}$ term, which requires evaluating the outcome of the remaining $k$-1 steps.
>
> Proposition 1 (summarized in Eq. 13) provides the key theoretical insight: it proves that the expected Q-value of a single-step denoised action $E_{\hat{p_k}}[Q]$ serves as a lower bound for the expected Q-value of the final, fully-refined action $E_{p_0}[Q]$.
>
> Therefore, by substituting the intractable multi-step expectation term in Eq. 12 with this simpler, single-step lower bound from Eq. 13, we directly arrive at the tractable surrogate objective presented in Eq. 14.
>
> **The key interpretation of using this lower bound is that** it enables us to effectively improve a complex multi-step process by optimizing a simpler single-step process. As long as we can continuously enhance the expected Q-value of single-step denoised actions $E_{\hat{p_k}}[Q]$, we are effectively optimizing a lower bound on the true objective, which in turn helps to improve the expected Q-value of the final (K-step) actions $E_{p_0}[Q]$.
>
> Based on this understanding, we derive our final loss function in Eq. 15.
>
> We have also revised the entire methodology section based on your suggestions, incorporating more explanations and discussions throughout.

---

> > ### Author Response · Authors · 2025-08-29
> >
> > **Q: There are different ways and losses to learn a distributional critic that has been used for RL before, such as in [1,2]; the paper lacks discussion on those other options.**
> >
> > ---
> >
> > Thank you for this insightful comment and for pointing us to these highly relevant and recent papers. We agree that there are various effective methods for learning a distributional critic, and we have revised our paper to better contextualize our approach within this active area of research. As you noted, recent works [1,2] provide compelling evidence for the benefits of moving away from traditional MSE regression, instead advocating for classification-style objectives to learn value distributions.
> >
> > In developing our approach, we focused on creating a stable learning system that would effectively support our novel policy improvement objective (Eq. 15). To this end, and given that this research was developed over the past 1.5 years, we chose to build upon the well-established C51 framework, augmented with the variance-reduction properties of clipped double Q-learning. When our work commenced, the alternative approaches [1,2] were not yet established, making this combination a natural choice due to its proven stability and effectiveness. This foundation allowed us to rigorously evaluate our primary technical contribution: the new actor update mechanism. To further validate the components of our critic, we have added a new ablation study in the ablation study part that demonstrates the additive performance gains from both the categorical distribution and the clipped Q-learning technique. The results further validate our choice, as the resulting system is highly performant and robust across various domains without requiring extensive, task-specific tuning.
> >
> > As suggested, we have revised our introduction to include a discussion of these recent approaches, situating our work within this broader context. We also discuss the potential application of these methods in our future work section. Thank you again for prompting this important clarification.
> >
> > ---
> >
> > [1] Imani, Ehsan, et al. "Investigating the histogram loss in regression." arXiv preprint arXiv:2402.13425 (2024).
> >
> > [2] Farebrother, Jesse, et al. "Stop regressing: Training value functions via classification for scalable deep RL." arXiv preprint arXiv:2403.03950 (2024).

---

### Review · Reviewer_Dmak · 2025-08-18

**Summary Of Contributions:**

The paper develops an actor-critic algorithm using a distributional critic and diffusion-based actor. The novelties of the approach include:
- Applying the idea of clipped double Q-learning to the distributional setting
- Applying the monotonic improvement theorem of TRPO to the training of the diffusion-based actor

**Audience:**

Yes

**Broader Impact Concerns:**

There is no Broader Impact statement. However, given this type of work, I believe this is fine.

**Claims And Evidence:**

No

**Requested Changes:**

I believe the paper would be strengthened as follows:
- The connection between the proposed method and model-based RL method, mentioned in the conclusion, should be explained in more details.
- The related work on model-free RL could be more comprehensive to include the baselines used in the experiments (e.g., SAC, DDPG), but also other classic algorithms (e.g., TD3). In particular, why use DDPG instead TD3, which is, I believe, a stronger baseline and also includes a mechanism to reduce overestimation?
- The notations should be explained the first time they appear(e.g., Eq. 7).
- I would suggest the authors to be more pedagogical and also recall the intuitions of the equations.
- More discussion of Proposition 1 is needed.
- In the experimental part, the authors should discuss hyperparameter tuning (if any) and computational times (with relevant information of compute specifications).
- An ablation study is needed to confirm the usefulness of the distributional critic with the clipped double Q-learning mechanism.
- In general, the paper should be more polished, since there are quite a few typos (e.g., "opinated"; "where" after equations doesn't need capitalization; "give us give us"; "equation equation"…)

**Strengths And Weaknesses:**

**Pros:**

The design of the actor-critic scheme is based on a novel application and combination of existing techniques.

The authors demonstrate that the proposed algorithm has good performance on a range of RL tasks and has nice properties such that yielding policies that induce more diverse trajectories.

**Cons:**

I believe that some statements do not follow from the experimental results. For instance:
- The last sentence of paragraph 3 in the introduction is not empirically verified. Unless I missed it, there's no ablation study where the distributional critic is changed to a point-estimate critic, or where the clipped double Q-learning mechanism is removed.
- How is the claim of the last paragraph of page 8 suggested by the experiments?

The theoretical result presented in Proposition 1 seems strange to me. Notably, assumption (ii) doesn't make sense to me (how can larger variance leads to smaller KL divergence?). Isn't it the reverse? Also, it's not clear how assumption (iii) can always hold in practice.

The proposed algorithm is probably more computationally costly than other model-free algorithms. However, this point is not discussed in the paper.

The presentation of the experimental results should be complemented with a discussion on hyperparameter tuning (if any) and a discussion on computational times (with relevant information of compute specifications).

The writing could be improved to make it easier to read for a general audience and could be more polished to remove typos.

---

> ### Author Response · Authors · 2025-08-29
>
> **Q: The last sentence of paragraph 3 in the introduction is not empirically verified. Unless I missed it, there's no ablation study where the distributional critic is changed to a point-estimate critic, or where the clipped double Q-learning mechanism is removed.**
>
> ---
>
> We thank the reviewer for their valuable feedback. We agree that our original manuscript lacked a direct ablation of the distributional critic and clipped double Q-learning.
>
> To directly address your concern, we have conducted a new ablation study in the paper. In this new experiment, we directly isolate the contributions of our critic's key components by comparing the full D2AC against versions where we either replace the distributional critic with a scalar one or remove the clipped double Q-learning mechanism. The results provide clear empirical validation: removing either of these components leads to a significant degradation in performance, confirming that their synergy is critical to D2AC's success.
>
> ---
>
> **Q: How is the claim of the last paragraph of page 8 suggested by the experiments? The connection between the proposed method and model-based RL method, mentioned in the conclusion, should be explained in more details.**
>
> ---
>
> Thank you for this insightful question about our work's motivation.
>
> We acknowledge this is more of an intuitive hypothesis rather than a definitive claim. Our experimental results show that D2AC successfully closes the performance gap with state-of-the-art model-based TD-MPC2, while the Two-hot + CDQ baseline (representing TD-MPC2's model-free component) performs relatively poorly (as shown in Figures 3 and 4). This suggests TD-MPC2 may rely heavily on its planning module for its strong performance.
>
> D2AC achieves comparable results through a purely model-free approach by incorporating two principles that parallel those in modern planners: (1) modeling a distribution of returns, and (2) iteratively refining action proposals.
>
> While this doesn't establish a formal equivalence, it suggests that certain benefits traditionally associated with planning-based methods might be captured within a well-designed model-free framework. To better articulate this connection, we have revised the discussions in both the Related Work and Experiments sections, more explicitly framing our claim as a hypothesis that is tested and supported by our experimental results.

---

> > ### Author Response · Authors · 2025-08-29
> >
> > **Q: The theoretical result presented in Proposition 1 seems strange to me. Notably, assumption (ii) doesn't make sense to me (how can a larger variance lead to smaller KL divergence?). Isn't it the reverse? Also, it's not clear how assumption (iii) can always hold in practice.**
> >
> > ---
> >
> > We sincerely thank the reviewer for this insightful question and for carefully examining the theoretical underpinnings of our work.
> >
> > **Regarding Assumption (ii):**
> >
> > You are absolutely correct. Our initial description of the KL divergence as ''decreasing'' was a typo. It should indeed be ''non-decreasing'', as a higher noise variance should not bring the policy closer to the optimal one.
> >
> > We have corrected this in the revised manuscript and the corresponding proof in the appendix, and are grateful for this crucial correction.
> >
> > **Regarding Assumption (iii):**
> >
> > We thank the reviewer for this critical question. This is indeed a mild assumption in our derivation. In practice, two design choices in our implementation make this assumption reasonable:
> >
> > First, we explicitly bound the logarithm of the learnable output standard deviation, preventing the noise term from collapsing to zero. Second, an entropy maximization term in our policy objective further encourages a non-trivial level of output noise.
> >
> > These mechanisms ensure the single-step $\hat{p}_0$ maintains randomness compared to the fully-refined $p_0$, making Assumption (iii) a reasonable precondition for our framework.
> >
> > ---
> >
> > **Q: The proposed algorithm is probably more computationally costly than other model-free algorithms. However, this point is not discussed in the paper.**
> >
> > ---
> >
> > We agree that a discussion on the computational cost of D2AC is crucial. To address this, we have added Appendix A: Analysis of Computational Cost, which compares the wall-clock runtime of D2AC against both SAC and TD-MPC2.
> >
> > You are correct: when looking purely at optimization steps per second, SAC is faster than D2AC. However, our analysis shows that this speed does not lead to success on challenging tasks. The core finding is that D²AC makes a deliberate trade-off: it accepts a modest increase in computational overhead per step to achieve the superior sample efficiency required to solve complex problems where simpler methods like SAC fail.
> >
> > ---
> >
> > **Q: The presentation of the experimental results should be complemented with a discussion on hyperparameter tuning (if any).**
> >
> > ---
> >
> > Thank you for this suggestion. We have expanded the Appendix to include our hyperparameter tuning strategy.
> >
> > In this discussion, we clarify that D2AC is highly robust and does not require extensive per-task tuning. Our results were achieved using a single set of standard hyperparameters across each environment family. The primary exception was setting the $V_{min}$ and $V_{max}$ values for the distributional critic, which we adjusted based on the reward scale of each family of environments (e.g., DM Control vs. Multi-Goal RL), as detailed in the Appendix. This robustness to hyperparameters represents another advantage of our method.

---

> > > ### Author Response · Authors · 2025-08-29
> > >
> > > **Q: The related work on model-free RL could be more comprehensive to include the baselines used in the experiments (e.g., SAC, DDPG), but also other classic algorithms (e.g., TD3). In particular, why use DDPG instead TD3, which is, I believe, a stronger baseline and also includes a mechanism to reduce overestimation?**
> > >
> > > ---
> > >
> > > Regarding the baselines for the goal-conditioned tasks, we used DDPG+HER for direct comparability with the canonical literature, as DDPG was the algorithm used in the original Hindsight Experience Replay paper [1]. While TD3 is a strong baseline, we opted to compare against more recent methods suited for goal-conditioned tasks like BVN [2] and distributional HER for a more rigorous benchmark. It's worth noting that our D2AC algorithm already incorporates TD3's key innovation—clipped double Q-learning—within our distributional critic design. This allows us to benefit from TD3's overestimation bias mitigation while extending it to the distributional setting.
> > >
> > > One key finding from our goal-based experiments is that D2AC serves as a strong base algorithm that works effectively out-of-the-box across various settings, from continuous dense-reward tasks to goal-conditioned environments.
> > >
> > > We appreciate the suggestion to expand our related work section and will ensure these connections are more clearly explained in the revised manuscript.
> > >
> > > ---
> > >
> > > - [1] Andrychowicz, Marcin, et al. "Hindsight experience replay." Advances in neural information processing systems 30 (2017).
> > >
> > > - [2] Hong, Zhang-Wei, Ge Yang, and Pulkit Agrawal. "Bilinear value networks." arXiv preprint arXiv:2204.13695 (2022).
> > >
> > > - [3] Eysenbach, Ben, Russ R. Salakhutdinov, and Sergey Levine. "Search on the replay buffer: Bridging planning and reinforcement learning." Advances in neural information processing systems 32 (2019).
> > >
> > > ---
> > >
> > > **About technical clarity and overall polish:**
> > >
> > > ---
> > >
> > > We sincerely thank the reviewer for such valuable and detailed feedback. We agree that the paper needed improvement in clarity and polish.
> > >
> > > We have thoroughly revised the manuscript based on all suggestions, clarifying notations, expanding the intuition behind our equations (especially Proposition 1), and correcting all identified typos and grammatical errors. The paper should now be more polished and easier to follow.

---

### Decision · Action_Editor_8aG7 · 2025-09-22

**Recommendation:** Accept as is

**Audience:**

Yes

**Audience Explanation:**

All three reviewers agree that the submission is of interest to part of the TMLR
readership. I share their opinion that it is of interest to RL researchers.

**Claims And Evidence:**

Yes

**Claims Explanation:**

While all reviewers agree that the work relies on known techniques,
two agree that their combination is still novel and interesting.
Reviewer Dmak points to concurrent work published on arXiv shortly after this submission.
Given the overlapping ideas, referencing it as concurrent work in the camera-ready
version of the submission will be useful to the readership (without necessitating any empirical comparisons).

---

> ### Author Response · Authors · 2025-10-03
>
> We sincerely thank the reviewers, Action Editor, and Editors-in-Chief for their thorough evaluation and guidance. The extensive, constructive feedback received during the review process has substantially enhanced the quality and rigor of our manuscript.
> We have uploaded the final camera-ready version and would like to express our gratitude to the TMLR community for their valuable support throughout this process.